# Sonar-TS: Search-Then-Verify Natural Language Querying for Time Series Databases

**Zhao Tan** [1 2]   **Yiji Zhao** [3]   **Shiyu Wang**   **Chang Xu** [4]   **Yuxuan Liang** [5]
**Xiping Liu** [† 1]   **Shirui Pan** [2]   **Ming Jin** [† 2]

## Abstract

Natural Language Querying for Time Series Databases (NLQ4TSDB) aims to assist non-expert users retrieve meaningful events, intervals, and summaries from massive temporal records. However, existing Text-to-SQL methods are not designed for continuous morphological intents such as shapes or anomalies, while time series models struggle to handle ultra-long histories. To address these challenges, we propose **Sonar-TS**, a neuro-symbolic framework that tackles NLQ4TSDB via a "Search-Then-Verify" pipeline. Analogous to active sonar, it utilizes a feature index to "ping" candidate windows via SQL, followed by generated Python programs to "lock on" and verify candidates against raw signals. To enable effective evaluation, we introduce **NLQTS-Bench**, the first large-scale benchmark designed for NLQ over TSDB-scale histories. Our experiments highlight the unique challenges within this domain and demonstrate that Sonar-TS effectively navigates complex temporal queries where traditional methods fail. This work presents the first systematic study of NLQ4TSDB, offering a general framework and evaluation standard to facilitate future research. Our code has been made available at https://github.com/Atlamtiz/Sonar-TS.

## 1. Introduction

The volume of time series data is increasing rapidly across domains such as IoT monitoring, financial trading, and AIOps. To handle this scale, specialized Time Series

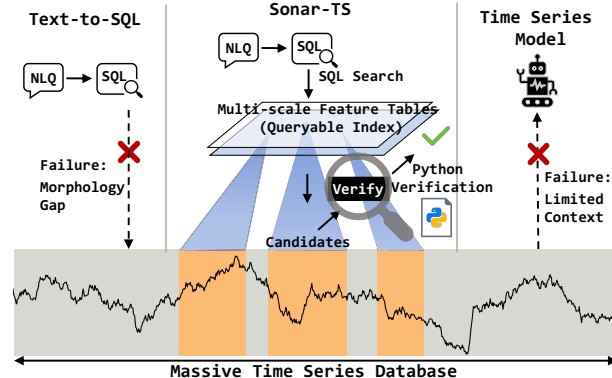

*Figure 1.* Comparison of querying paradigms. While Text-to-SQL fails to express morphological intents and Time Series Models are limited by context length, Sonar-TS adopts a "Search-Then-Verify" pipeline: it uses SQL to search a symbolic index for candidates and Python to verify them on raw data.

Databases (TSDBs) have become the standard solution for storage (Pelkonen et al., 2015). However, querying these massive records for meaningful insights remains a significant barrier for non-expert users. Unlike simple numerical lookups (e.g., "maximum value in May"), users often prioritize morphological characteristics, such as identifying a specific day where data shows "a rapid rise followed by a slow fall." The profound semantic gap between such abstract natural language descriptions and the continuous numerical data constitutes the fundamental challenge in TSDB querying.

Existing attempts to bridge this gap can be broadly categorized into two distinct paradigms. On the one hand, from a database-centric perspective, Text-to-SQL (Pourreza & Rafiei, 2023; Tan et al., 2024; Liu et al., 2025) aims to translate natural language into executable SQL queries. While this field has demonstrated significant success in handling complex schema linking within relational databases, it encounters an expressivity bottleneck in the time series domain: standard SQL lacks native primitives to describe continuous morphological concepts (e.g., shapes or trends), making it arduous to formulate qualitative intents using rigid operators. On the other hand, from a data-centric perspec-

---

[1]Jiangxi University of Finance and Economics [2]Griffith University [3]Yunnan University [4]Microsoft Research Asia [5]The Hong Kong University of Science and Technology (Guangzhou). Correspondence to: Xiping Liu <liuxiping@jxufe.edu.cn>, Ming Jin <mingjinedu@gmail.com>.

*Proceedings of the 43rd International Conference on Machine Learning*, Seoul, South Korea. PMLR 306, 2026. Copyright 2026 by the author(s).

tive, Time Series Question Answering (TSQA) (Jin et al., 2024; Langer et al., 2025; Divo et al., 2025) focuses on aligning textual modalities with raw temporal signals, enabling models to interpret morphological patterns directly. However, these models face a severe scalability bottleneck. Constrained by finite context windows, end-to-end models cannot ingest the ultra-long-horizon histories (often millions of points) stored in real-world TSDBs.

To overcome these limitations, we formally define the task of **Natural Language Querying for Time Series Databases (NLQ4TSDB)**. Unlike standard TSQA which typically operates on short context windows, this task demands that a system ground high-level semantic intents into executable operations over massive, unsegmented temporal records. To facilitate rigorous evaluation, we introduce **NLQTSBench**, a comprehensive benchmark featuring a hierarchical taxonomy spanning four levels of complexity: Level 1 tests basic numerical retrieval and windowing; Level 2 focuses on morphological pattern recognition (e.g., shapelets identification); Level 3 evaluates semantic reasoning over composite trends and causal anomalies; and Level 4 requires holistic insight synthesis for report generation.

In this work, we propose **Sonar-TS**, a neuro-symbolic framework tailored to address the unique challenges of NLQ4TSDB. As illustrated in Figure 1, Sonar-TS reframes the TSDB querying process as a "Search-Then-Verify" pipeline, drawing an analogy to active sonar. Instead of scanning the entire raw history (which is computationally prohibitive) or relying solely on SQL (which lacks morphological expressivity), our system first "pings" the massive search space using a multi-scale feature index via SQL queries to localize candidate windows. Subsequently, it "locks on" to these candidates using generated Python programs to verify the raw signals against the user's semantic intent. The framework is structured into three key modules: (1) *Offline Data Processing*, which constructs compact feature tables to render continuous shapes queryable; (2) *Online Querying*, where an LLM-driven planner synthesizes hybrid SQL-Python execution plans; and (3) *Post-processing*, which formats execution artifacts into user-friendly insights and visualizations.

Our contributions are summarized as follows:

- **New Problem:** We formally define the **NLQ4TSDB** task, highlighting its necessity in practice, the dual challenges of semantic grounding, and scalability that differentiate it from traditional Text-to-SQL and TSQA.

- **New Benchmark:** We introduce **NLQTSBench**, the first large-scale benchmark for complex time series question-answering on TSDBs. Distinguished from existing settings that are restricted to short context windows, it necessitates active evidence localization over

long-horizons rather than passive context processing.

- **Novel Framework:** We propose **Sonar-TS**, a framework that orchestrates a "Search-Then-Verify" workflow. Our experiments show that Sonar-TS handles complex temporal queries where traditional methods fail, laying a foundation for future research in DB-grounded time series analysis.

## 2. The NLQ4TSDB Problem

### 2.1. Problem Formulation

The goal of NLQ4TSDB is to retrieve insights from a Time Series Database (TSDB) via natural language. Formally, let $\mathcal{D}$ denote the TSDB instance containing massive timestamped records. Its structure is defined by a schema $\{M_i\}_{i=1}^n$, consisting of $n$ measurements (e.g., tables). We define each measurement as a tuple:

$$M_i = (n_i, \mathcal{T}_i, \mathcal{F}_i), \tag{1}$$

where $n_i$ is the measurement name, $\mathcal{T}_i$ is a set of tags (categorical), and $\mathcal{F}_i$ is a set of fields (numerical).

**Input.** The input consists of two components:

1. *Textual Schema* ($\mathcal{S}$), a serialized token sequence of the metadata $\{M_i\}_{i=1}^n$, formally defined as $\mathcal{S} = \text{Serialize}(\{M_i\}_{i=1}^n)$. This provides structural context without exposing the raw data in $\mathcal{D}$.

2. *Natural Language Query* ($Q$), a sentence specifying the user intent, typically targeting temporal patterns rather than simple relational lookups.

**Objective.** We formulate the task as a two-stage process. A solver $f$ first synthesizes an executable query plan $\pi$ (e.g., SQL or Python code) based on the schema, which is then executed against the instance $\mathcal{D}$ to yield the answer $A$:

$$\pi = f(Q, \mathcal{S}), \qquad A = \text{Exec}(\pi, \mathcal{D}). \tag{2}$$

**Output.** The answer $A$ can be a scalar, timestamps, time intervals, or descriptive text, depending on the query intent.

### 2.2. Challenges

To illustrate the unique challenges of NLQ4TSDB, we consider a representative query targeting a composite trend:

> NLQ: "Identify the specific dates within the last year at which the temperature showed a rapid increase and then maintained a stable plateau."

This intent is easy for a human to understand. However, executing it over a massive TSDB in practice exposes three core challenges.

**C1: The Representation Gap.** The query relies on geometric shapes such as "rapid rise" and "stable plateau". A TSDB, in contrast, only stores point-by-point numerical values. SQL is built on strict boolean logic with fixed thresholds, so it cannot directly express sequential shapes. For example, defining a "rapid rise" as `WHERE slope > 60` is fragile, because the meaning of a shape depends on its context and varies across series. A clear gap therefore exists between the user's shape-based intent and the point-based storage of a TSDB. Closing this gap requires an abstraction layer that turns continuous shapes into a discrete, queryable form the database can search.

**C2: The Context Scale Limit.** Recent time series multimodal models (Xie et al., 2025) can interpret shapes well, but they do not match the scale of real TSDBs. A query over "the last year" may cover millions of raw points, which far exceeds the context window of modern models. Scanning the full history with a sliding window is also too slow for online use. As a result, these models still cannot directly handle TSDB-scale histories.

**C3: The Semantic Grounding Conflict.** Natural language is inherently fuzzy. Users describe patterns with relative terms such as "rapid rise" or "gradual drop". Database search, however, needs exact numerical parameters. When users think at this abstract level, they usually cannot give such numbers. The system must therefore translate the fuzzy words into concrete thresholds before execution. Using static thresholds is not enough, because the meaning of "rapid" depends on the local data distribution and the historical context. Resolving this conflict requires a method that maps vague descriptions into precise and context-aware numerical boundaries.

## 3. Related Work

NLQ4TSDB is a novel problem closely related to three fields: Text-to-SQL, Time Series Question Answering (TSQA), and Time Series Similarity Search. However, solving NLQ4TSDB task requires three capabilities at once that no single field provides: Morphological Primitives (MP) to process continuous shapes, Massive Scalability (MS) to handle long database horizons, and Natural Language Grounding (NLG) to understand user intents. As shown in Table 1, each prior field covers only a subset of these capabilities, while Sonar-TS unifies all three. We discuss the specific limitations of each field below.

**Text-to-SQL.** Text-to-SQL translates natural language into executable SQL (Li et al., 2023; Qin et al., 2022), evolving from rule-based methods to LLM-driven agentic pipelines. Recent work adopts decomposition and multi-agent strategies to scale up. For example, DIN-SQL (Pourreza & Rafiei, 2023) and MAC-SQL (Wang et al., 2025a) split the task

*Table 1.* Capability comparison of existing paradigms against the fundamental requirements of NLQ4TSDB. Abbreviations: MP (Morphological Primitives), MS (Massive Scalability), NLG (Natural Language Grounding).

| Paradigm / Method | MP | MS | NLG |
|---|---|---|---|
| Text-to-SQL | ✗ | ✓ | ✓ |
| TSQA Models | ✓ | ✗ | ✓ |
| TS Similarity Search | ✓ | ✓ | ✗ |
| **Sonar-TS (Ours)** | ✓ | ✓ | ✓ |

into sub-problems such as schema linking and results refinement; CHASE-SQL (Pourreza et al., 2025) verifies candidate queries via test-time computation; and CHESS (Talaei et al., 2024) uses hierarchical retrieval to filter irrelevant schema items at industrial scale. However, Text-to-SQL is built on relational algebra and lacks native primitives for shape matching. Concepts like "V-shape" or "fluctuation" cannot be expressed in plain SQL, leaving a semantic gap that pure SQL generation cannot close.

**Time Series Question Answering.** TSQA enables models to perceive and reason over numerical signals, moving beyond forecasting toward semantic understanding (Chang et al., 2025). Time-LLM (Jin et al., 2024) reprograms LLMs by mapping series into textual prototypes, while ChatTS (Xie et al., 2025) aligns signals with language via multimodal encoders. Benchmarks such as Time-MQA (Kong et al., 2025) and QuAnTS (Divo et al., 2025) further expand the scope of reasoning tasks. However, current TSQA models are bounded by the Transformer context window, which makes them infeasible for scanning massive, unsegmented histories in real TSDBs. For example, a year of high-frequency monitoring easily contains millions of points, far exceeding the few-thousand-token contexts typical of TSQA models.

**Time Series Similarity Search.** This field aims to find subsequences that match a given query sequence. Existing approaches fall into two broad families: index-accelerated numerical matching, and symbolic compression. For massive datasets, KV-Match (Wu et al., 2019) builds on a custom key-value index, while MS-Index (d'Hondt et al., 2025) uses R-trees over Discrete Fourier Transform approximations for efficient Top-$k$ retrieval. Symbolic methods like SAX (Lin et al., 2007) discretize continuous waveforms into discrete string tokens. These techniques are efficient, but they follow a "Query-by-Example" paradigm and require a precise numerical sequence as input. NLQ4TSDB instead follows "Query-by-Language", where users express intents through abstract linguistic concepts (e.g., "rapid decline"). This shift from a concrete numerical exemplar to an abstract textual description fundamentally changes the retrieval problem, making existing similarity search indices inapplicable.

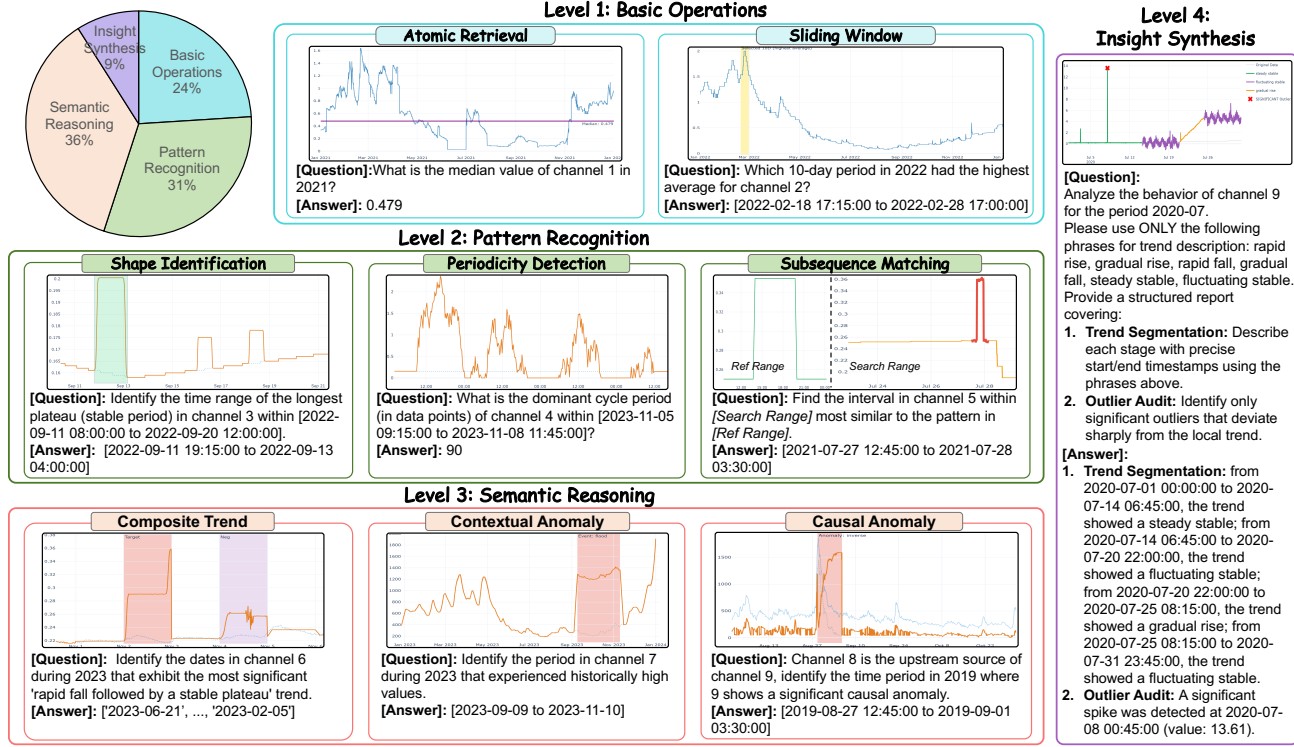

Figure 2. The hierarchical taxonomy of tasks in NLQTSBench. The benchmark ranges from **Level 1 (Basic Operations)** which tests numerical filtering, to **Level 2 (Pattern Recognition)** for morphological grounding, **Level 3 (Semantic Reasoning)** for logical composition, and finally **Level 4 (Insight Synthesis)** for narrative reporting.

## 4. NLQTSBench

NLQ4TSDB is a fundamentally new problem. The closest existing proxy is TSQA benchmarks (Kong et al., 2025; Chen et al., 2025), but they typically restrict evaluation to short snippets (often fewer than 500 points). This setting implicitly assumes that the relevant evidence has already been perfectly segmented, which misrepresents practical TSDB interactions where users query continuously accumulated, unsegmented histories.

To bridge this gap, we introduce **NLQTSBench**, the first benchmark designed specifically for the NLQ4TSDB problem. In NLQTSBench, the average search space per query spans approximately 12,000 points. This magnitude enforces a paradigm shift from passive context reading to active, long-horizon evidence localization.

### 4.1. Task Taxonomy

We organize NLQTSBench into a four-level taxonomy, illustrated in Figure 2. The hierarchy progresses from atomic numerical operations to holistic reasoning, evaluating how well a solver grounds abstract linguistic concepts into continuous time series at scale.

**Level 1: Basic Operations.** Tests precision in translat-

ing natural language constraints into executable numerical filters. It includes (1) *Atomic Retrieval*, covering statistical calculation, precise localization, and value-based range selection; and (2) *Sliding Window*, which identifies target intervals across a massive search space.

**Level 2: Pattern Recognition.** Assesses the ability to ground abstract morphological concepts (e.g., V-shape) into continuous numerical series. It includes (1) *Shape Identification*, recognizing qualitative patterns such as plateaus or spikes; (2) *Periodicity Detection*, capturing the dominant cycle in oscillating data; and (3) *Subsequence Matching*, retrieving periods that morphologically resemble a reference pattern.

**Level 3: Semantic Reasoning.** Evaluates logical composition and relational reasoning over temporal sequences and inter-channel dependencies. It includes (1) *Composite Trend*, handling multi-stage transitions such as a rapid rise followed by a slow fall; (2) *Contextual Anomaly*, diagnosing abnormalities relative to local history rather than fixed thresholds; and (3) *Causal Anomaly*, identifying contradictions between correlated series.

**Level 4: Insight Synthesis.** Serves as a composite evaluation, requiring the solver to orchestrate lower-level capabilities into a cohesive *Report Generation* workflow that

*Table 2.* Statistics of NLQTSBench. The dataset comprises 1153 queries spanning four complexity levels.

| Level | Task Sub-Type | Count | Avg. Length |
|---|---|---|---|
| **L1: Basic Operations** | Atomic Retrieval
Sliding Window | 216
58 | $\sim$ 2k |
| **L2: Pattern Recognition** | Shape Identification
Periodicity Detection
Subsequence Matching | 129
120
111 | $\sim$ 0.6k |
| **L3: Semantic Reasoning** | Composite Trend
Contextual Anomaly
Causal Anomaly | 220
104
95 | $\sim$ 30k |
| **L4: Insight** | Report Generation | 100 | $\sim$ 3k |
| **Total** | **All Tasks** | **1153** | **Avg. $\approx$ 12k** |

sequentially performs trend segmentation and outlier auditing within a localized window.

Table 2 summarizes the query distribution and context scales across the four levels. Full task definitions and query templates are provided in Appendix A.1.

### 4.2. Data Construction Pipeline

A NLQ4TSDB benchmark requires three ingredients: *massive raw histories, natural language queries, and precise ground truths.* The first two are straightforward, but ground-truth annotation is the bottleneck. Pinpointing a morphological boundary across years of data (e.g., the exact moment a "gradual rise" begins) is highly subjective and prohibitively labor-intensive for human annotators. No existing automated algorithm extracts such boundaries reliably from real-world data either.

We therefore adopt a controlled injection strategy: we synthesize mathematical patterns that strictly match the linguistic descriptions and overlay them onto real backgrounds sampled from CausalRivers (Stein et al., 2025), ETTm1 (Zhou et al., 2021), and SMD (Su et al., 2019). This makes labels auditable by construction while preserving the noise and drift of real signals, and follows a paradigm already adopted in recent TSQA work (Xie et al., 2025). The pipeline runs in three stages: template formulation, signal injection, and human visual verification. Full details are in Appendix A.2.

### 4.3. Benchmark Variants.

NLQTSBench is designed for long-horizon querying, but existing TSQA models often operate within small context windows and cannot process such long histories. To enable a fair comparison with these baselines, we release **NLQTSBench-Lite**, a compatibility set of 500 samples with fixed 512-point windows, covering three core capabilities: *Shape Identification*, *Composite Trend*, and *Causal Anomaly*. We emphasize that NLQTSBench-Lite is solely intended for TSQA baseline comparisons.

## 5. The Sonar-TS Framework

### 5.1. Overview

No existing paradigm covers all the capabilities required by NLQ4TSDB task. We therefore propose **Sonar-TS**, the first framework purpose-built for this task. As illustrated in Figure 3, the workflow operates in three stages:

- **Offline Data Processing** (Section 5.2). We preprocess native time series into multi-scale Feature Tables that act as a queryable semantic index. These tables store window-level metadata, statistical primitives, and morphological tokens to facilitate rapid and approximate evidence localization.

- **Online Querying** (Section 5.3). Given a query and the textual database schema, an LLM first plans the task and then generates hybrid SQL and Python programs. The SQL filters candidate windows from the feature tables, while the Python programs verify them against raw slices. This generation is supported by an offline *Prompt Cold Start* mechanism, which injects a static set of distilled *Experiences* to guide the LLM with domain-specific heuristics.

- **Post-processing** (Section 5.4). This step formats the raw execution artifacts into user-friendly responses, and provides lightweight visualizations for quick human verification.

### 5.2. Offline Data Processing

To avoid costly full scans at query time, we precompute compact multi-scale *Feature Tables* that act as a *Queryable Semantic Index*. The raw TSDB remains the definitive source of truth and is accessed only for exact computations on localized regions.

For each numeric channel, we materialize feature rows under a hierarchical windowing scheme across multiple granularities (e.g., Year/Month/Day). Each row corresponds to one time window and stores its metadata (e.g., Channel ID) along with two categories of descriptors that jointly support numerical and morphological queries.

1. **Statistical Primitives.** We compute lightweight statistics (e.g., `slope`, `std_val`) as cheap pruning signals. This lets the system prioritize candidate windows with simple SQL (e.g., `ORDER BY std_val DESC LIMIT 1`) instead of aggregating over raw points at query time.

2. **Morphological Tokens.** To accommodate shape-driven intents (e.g., "V-shape"), we must transform

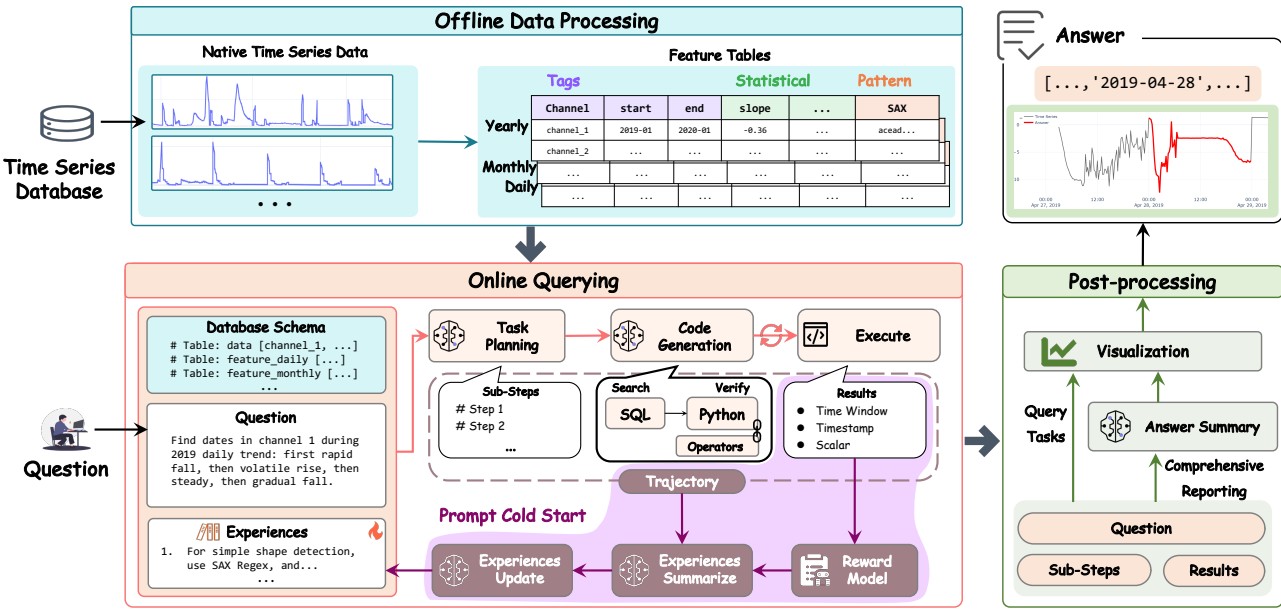

*Figure 3.* The overview of the **Sonar-TS** framework. The workflow is organized into three stages: (1) **Offline Data Processing** constructs compact multi-scale Feature Tables to serve as a queryable index; (2) **Online Querying**, where the Task Planner and Code Generator synthesize SQL for rapid candidate search and Python for exact verification, guided by domain heuristics distilled from an offline Prompt Cold Start mechanism; and (3) **Post-processing** translates execution artifacts into a user-friendly interface.

continuous numerical shapes into discrete string tokens. This symbolization is a strict necessity for enabling standard SQL engines to search massive time series data. We adopt Symbolic Aggregate approXimation (SAX) (Lin et al., 2007) as a well-established baseline, providing a transparent and reproducible starting point for Sonar-TS. We treat SAX as a *Symbolic Search Handle*, allowing the system to approximate shapes via standard SQL regex (e.g., approximating a "monotonic rise" via `WHERE regexp_like(sax, '[ab]+.*[de]+')`). Crucially, SAX is one valid instantiation rather than a hard dependency: our framework is agnostic to the tokenization method, and any advanced discrete representation can be integrated.

### 5.3. Online Querying

Querying over TSDBs faces two core challenges: (1) *Computational Infeasibility*: Verifying these intents directly on raw data requires full-history scanning, which is prohibitively expensive for online interaction. (2) *The Semantic-Signal Mismatch*: Users' intents often describe high-level morphology (e.g., "steady → sharp drop") but databases only index low-level raw signals. Expressing such composite intents as a single SQL predicate is often impossible.

To resolve these issues, Sonar-TS adopts a Search-Then-Verify workflow that orchestrates explicit collaboration between coarse-grained symbolic database search and fine-grained algorithmic verification.

**Search-Then-Verify Workflow.** The pipeline takes the user query, the database schema (including offline feature tables), and the Experience set as inputs, and proceeds in three sequential steps.

**Step 1: Task Planning.** Complex temporal queries are inherently difficult to execute in a single shot. We employ the backbone LLM as a Task Planner that decomposes the query into an ordered sequence of fine-grained sub-steps, identifying both the logical execution order and the operator types required for each step.

**Step 2: Code Generation.** Based on the plan, the Code Generator instantiates the abstract logic into executable forms via two distinct mechanisms:

1. **SQL-based Search:** To narrow the massive search space, the Generator writes SQL targeting the offline feature tables using SAX tokens as symbolic handles. For instance, to search for a composite trend like "rapid rise, then fall," it formulates a fuzzy regex (e.g., `WHERE regexp_like(sax, '[ab]+.*[de]+.*[ab]+')`) to retrieve approximate candidate windows. This strategy efficiently eliminates structurally irrelevant histories, yielding a manageable set of candidate metadata.

2. **Operator-based Verification:** Because SAX compression inevitably introduces information loss, the localized candidates must be rigorously validated. The Generator synthesizes Python programs that fetch the

*Table 3.* Representative verification primitives in the operator library.

| Operator Signature | Mathematical Grounding |
|---|---|
| detect_period | Autocorrelation Function (ACF) for cyclic behaviors. |
| find_best_match | Dynamic Time Warping (DTW) for subsequence matching. |
| detect_changepoints | PELT algorithm for structural segmentation. |
| calc_trend_slope | Theil-Sen estimator + local slope distribution. |
| calc_correlation | Pearson correlation for causal links. |

raw data slices corresponding to these candidates and execute exact mathematical checks. Continuing our example, the script applies calc_trend_slope and detect_changepoints to the raw slices. It verifies a "rapid" rise by checking if the candidate's slope ranks within the top percentiles of the historical slope distribution. To support this, we equip the Generator with a standard library of classic time-series algorithms wrapped as executable operators. Table 3 highlights representative operators used for verification; the comprehensive library and implementation details are provided in our open-source repository. This design ensures that fuzzy linguistic concepts are strictly grounded in established, context-aware mathematics.

**Step 3: Execution & Self-Correction.** The generated SQL and Python codes are executed sequentially. This stage handles runtime failures (e.g., SQL returning empty results or Python syntax errors). If an error occurs, the execution feedback (a traceback or a result summary) is returned to the Code Generator for refinement, forming a closed-loop debugging process that iterates until successful execution or a predefined retry limit is reached.

**Prompt Cold Start.** NLQ4TSDB is knowledge-intensive, requiring domain heuristics. To inject such expertise without fine-tuning, Sonar-TS maintains Experiences: compact, high-level insights distilled from past executions (e.g., correct operator usage, effective SAX matching, and avoidance of irregular sampling gaps). At inference, the current Experience set is injected directly into the Planner and Generator prompts as a concise "expert manual."

Crucially, Experiences are built entirely offline on a profiling dataset and never updated at query time, avoiding both fine-tuning cost and the unpredictability of online memory updates. The construction follows a closed-loop "Reward-Summarize-Update" cycle: first scores each execution; an Experience Summarizer distills 1–3 skills from the execution trajectory (plan, code, reward); and an Experience Updater refines the global set via add/merge/delete operations, enforcing a strict cap (e.g., 20 insights) to bound context overhead. Figure 4 shows representative entries, ranging from high-level algorithmic orchestration to low-level syntax corrections, each a targeted heuristic rather than generic prompt advice.

```
Snapshot of Injected Experiences
• For symbolic pattern matching, utilize the 5-level SAX alphabet ('a-b' for
  Low, 'd-e' for High). E.g., to detect a 'low plateau' (bottom out), use the
  fuzzy SQL regex: SELECT * FROM feature_daily WHERE sax REGEXP '.*[a-b]{3,}.*'
• When working with timestamps in pandas, ALWAYS convert them immediately after
  loading: df['ts'] = pd.to_datetime(df['ts']). Never assume they are datetime
  objects. Never call .strftime() on a numpy array without conversion.
• For structural trend reporting, do not calculate a global slope. Instead, use
  a divide-and-conquer approach: first apply detect_changepoints to decompose
  the trend into segments, then use calc_trend_slope to quantify the direction
  of each specific segment.
• If Step 1 (SAX filtering) returns an empty result, you MUST automatically
  fallback to fetching raw data for the entire requested time range. Do not
  simply return 'No data found'—the SAX approximation has been too strict.
• When detecting continuous time periods (Interval Discovery), use dynamic gap
  detection based on the dataset's median sampling interval.
• ... (others)
```

*Figure 4.* Snapshot of representative Injected Experiences.

### 5.4. Post-processing

The Online Querying module returns raw mathematical outputs (e.g., timestamps, intervals, or scalars) that are not always directly consumable by users. The Post-processing module wraps these outputs into user-facing answers. For text-centric tasks such as Insight Synthesis, an LLM is invoked to compose a concise summary that integrates the retrieved values into a coherent narrative. In addition, Post-processing optionally produces a lightweight Python visualization that renders the relevant time-series context and highlights the retrieved windows, supporting quick visual inspection and human verification.

### 5.5. Execution Complexity

Sonar-TS delegates heavy computation to the database engine and Python operators. Let $R$ be the retry limit, $T_{\mathrm{LLM}}$ the LLM call cost, $M$ the feature-table rows scanned, $K$ the retrieved candidate windows, and $w$ the window length. The worst-case online cost is:

$$O\big(R \cdot T_{\mathrm{LLM}} + M + K \cdot f(w)\big), \tag{3}$$

where $f(w)$ is the operator complexity on one window. The key property is decoupling from the series length $N$: the LLM prompt depends only on the schema and a bounded Experience set, and Python verification depends only on $K$. The term $M$ scales with the feature tables, not raw data, and standard indices keep it efficient. Sonar-TS therefore avoids the $O(N)$ or $O(N^2)$ scans that limit end-to-end time series models on long histories.

### 5.6. Implementation Details

Sonar-TS is a training-free framework targeting SQL-compatible TSDBs, with DeepSeek-V3 as the default backbone. Full implementation details, including feature table and hyperparameter configurations, the experience initialization protocol, and all prompt templates, are provided in Appendix B. The complete configuration files and operator library are released in our open-source code.

*Table 4.* Main experimental results on NLQTSBench. Abbreviations: **AR** (Atomic Retrieval), **SW** (Sliding Window), **SI** (Shape Ident.), **PD** (Periodicity Det.), **SM** (Subseq. Matching), **CT** (Composite Trend), **CxA** (Contextual Anomaly), **CsA** (Causal Anomaly), **IS** (Insight Synthesis). Best results are **bolded**, and the best baseline results are underlined.

| Category | Method | L1: Basic Ops | | L2: Pattern Rec. | | | L3: Semantic Reasoning | | | L4 | Avg. |
|---|---|---|---|---|---|---|---|---|---|---|---|
| | | AR | SW | SI | PD | SM | CT | CxA | CsA | IS | |
| *Performance on NLQTSBench-Lite (Short Context)* | | | | | | | | | | | |
| Time Series Models | ChatTS-14B | - | - | 0.1768 | - | - | 0.2431 | - | 0.1229 | - | 0.1818 |
| | ITFormer-7B | - | - | 0.0736 | - | - | 0.1500 | - | 0.1953 | - | 0.1529 |
| | Time-R1 | - | - | 0.0320 | - | - | 0.1395 | - | 0.1878 | - | 0.1374 |
| **Ours** | **Sonar-TS** | - | - | **0.2491** | - | - | **0.2680** | - | **0.3615** | - | **0.3016** |
| *Performance on NLQTSBench (Long History)* | | | | | | | | | | | |
| Text-to-SQL Methods | MAC-SQL | 0.4735 | 0.0457 | 0.0419 | 0.3928 | 0.0598 | 0.0020 | 0.0346 | 0.0152 | - | 0.1611 |
| | Xiyan-SQL-32B | 0.1082 | 0 | 0.0588 | 0.2263 | 0.0068 | 0.0021 | 0.0168 | 0.0020 | - | 0.0582 |
| | Omni-SQL-32B | 0.4245 | 0.0086 | 0.0185 | 0 | 0 | 0.0038 | 0.0154 | 0 | - | 0.0921 |
| **Ours** | **Sonar-TS** | **0.8609** | **0.7827** | **0.3336** | **0.8625** | **0.9439** | **0.2988** | **0.4767** | **0.3841** | **0.7395** | **0.6144** |

## 6. Experiments

### 6.1. Experimental Setup

**Datasets.** No prior benchmark exists for NLQ4TSDB; we therefore evaluate on our own NLQTSBench. For fair comparison across different baselines, we adopt a dual-track setup: the full benchmark (1,153 queries over massive histories) is used for query-based methods, while NLQTSBench-Lite (500 queries with 512-point windows) accommodates context-limited end-to-end time series models.

**Baselines.** NLQ4TSDB sits at the intersection of Text-to-SQL and time series models, with no direct baseline available; we therefore compare against representatives from both sides.

- **Time series models:** ChatTS (Xie et al., 2025), a multimodal framework aligning signals with text features; ITFormer (Wang et al., 2025b), a Transformer-based encoder for temporal modeling; and Time-R1 (Luo et al., 2025), a reasoning-enhanced time series model.

- **Text-to-SQL methods:** MAC-SQL (Wang et al., 2025a), a classic multi-agent decomposition framework; Xiyan-SQL (Liu et al., 2026) and Omni-SQL (Li et al., 2025), representing leading end-to-end SQL generation models.

**Evaluation Metrics.** NLQTSBench includes nine distinct tasks with diverse output formats, so we adopt format-specific metrics: IoU for time intervals (e.g., Shape Identification), accuracy for scalars and timestamps (e.g., Atomic Retrieval), F1-score for date sets (e.g., Composite Trend), and a composite score for free-form reports (Insight Synthesis). Full definitions and the task-to-metric mapping are provided in Appendix A.3.

### 6.2. Main Results

Table 4 reports the comparative performance on NLQTS-Bench. The results confirm that NLQ4TSDB poses a unique challenge: it requires a synergy of precise morphological grounding and complex logical reasoning, which no existing baseline can effectively solve. Against this backdrop, Sonar-TS achieves consistent improvements across all categories, outperforming every baseline by a large margin.

**Comparison with Time Series Models (NLQTSBench-Lite).** Time series models perform poorly, even on tasks they are expected to solve well, such as Shape Identification (SI). Interestingly, they often score lower on simple SI tasks than on Composite Trend (CT). Qualitative analysis (Section 6.4) suggests that while these models can recognize general shapes, they struggle with strict semantic filtering: given a query to "Identify the longest plateau", the model effectively locates a plateau but ignores the "longest" constraint. Sonar-TS itself shows reduced accuracy on morphology tasks (SI, CT) in this short-context setting compared to the long-history benchmark, likely due to information loss from SAX compression on short windows.

**Comparison with Text-to-SQL Methods (NLQTSBench).** Within the Text-to-SQL category, the training-free MAC-SQL outperforms fine-tuned SOTA models like Xiyan-SQL and Omni-SQL. This trend runs contrary to general benchmarks such as BIRD (Li et al., 2023). We attribute this divergence to a fundamental paradigm gap: relational querying focuses on discrete record filtering, whereas time series analysis demands reasoning over continuous temporal patterns. SQL baselines therefore struggle on morphology-dependent tasks (e.g., SI, CT). For instance, to identify a "slow rise followed by rapid ascent", they typically generate static range constraints (e.g., `WHERE value > threshold`) that cannot capture the dynamic rate of change.

*Table 5.* Ablation study on component contributions. We analyze the impact of removing the Feature Table, Experiences, and the Verification phase. Abbreviations follow Table 4. Cells highlighted in blue indicate tasks most significantly impacted by the ablation.

| Variant | L1: Basic | | L2: Pattern | | | L3: Reasoning | | | L4 | Avg. |
|---|---|---|---|---|---|---|---|---|---|---|
| | AR | SW | SI | PD | SM | CT | CxA | CsA | IS | |
| **Sonar-TS** | **0.8609** | **0.7827** | **0.3336** | **0.8625** | **0.9439** | **0.2988** | **0.4767** | **0.3841** | **0.7395** | **0.6144** |
| w/o Self-Correction | 0.8302 | 0.7250 | 0.3056 | 0.8618 | 0.9217 | 0.2919 | 0.4813 | 0.3326 | 0.7139 | 0.5930 |
| w/o Feature Tables | 0.8658 | 0.6923 | 0.1561 | 0.8804 | 0.9309 | 0.0422 | 0.4968 | 0.3918 | 0.7163 | 0.5430 |
| w/o Experiences | 0.8426 | 0.5591 | 0.1186 | 0.8311 | 0.9138 | 0.0677 | 0.2501 | 0.3612 | 0.3415 | 0.4686 |
| w/o Verification | 0.8294 | 0.1870 | 0.2992 | 0.0154 | 0.0396 | 0.2826 | 0.0818 | 0.0592 | 0.0261 | 0.2721 |

## 6.3. Ablation Study

Table 5 reports the contribution of each Sonar-TS component when ablated. The four components produce qualitatively different failure modes, ranging from a localized robustness drop to near-total system collapse, as we now discuss in order of increasing impact.

**Impact of Self-Correction.** Removing the self-correction loop primarily affects SW tasks (0.78 → 0.73). Since SW relies on generated Python scripts, this loop is needed to recover from initial syntax or logic errors. The modest overall drop suggests it acts as a robustness refinement rather than a core capability.

**Impact of Feature Tables.** Without Feature Tables, morphology-driven tasks collapse: SI drops 0.33 → 0.16 and CT drops 0.30 → 0.04, while L1 tasks remain intact. This symbolic index is essential for shape retrieval. Notably, this ablation reduces Sonar-TS to a generic SQL+Python agent, whose collapse shows that agent-only paradigms cannot solve NLQ4TSDB without symbolic indexing.

**Impact of Experiences.** Without Experiences, reasoning-heavy tasks degrade broadly (CxA 0.48 → 0.25, IS 0.74 → 0.34), and morphology cues weaken too (SI, CT). This module bridges generic LLM logic and domain heuristics, letting the planner mimic expert analytical workflows.

**Impact of Verification.** Ablating Verification causes near-total collapse (Avg. 0.61 → 0.27), zeroing out algorithmic tasks such as PD (0.86 → 0.02) and SM (0.94 → 0.04). Verification is the indispensable computational engine for rigorous procedures that cannot be expressed in plain SQL.

## 6.4. Case Study

Figure 5 presents a qualitative analysis of the Shape Identification task, specifically targeting the "longest plateau". Text-to-SQL baselines struggle to express continuous morphological traits, often hallucinating rigid numerical filters that fail to capture the geometric pattern. Conversely, time series models correctly recognize typical plateau shapes within short contexts but fail to align with the "longest"

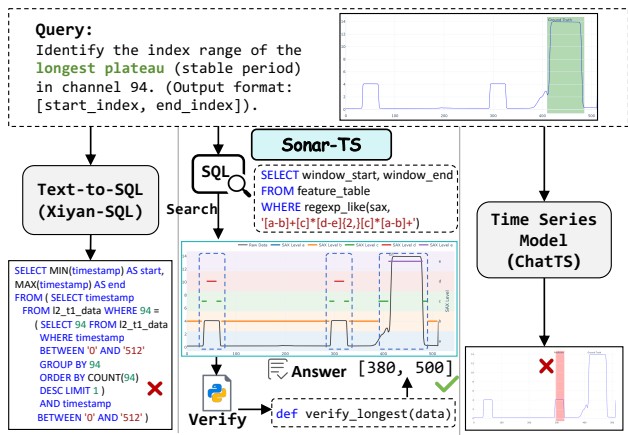

*Figure 5.* Case Study. Text-to-SQL (Left) lacks morphological expressivity, and TS Models (Right) fail the logical constraint. Sonar-TS (Middle) succeeds via Search-Then-Verify.

intent, lacking the global reasoning to compare durations. Sonar-TS bridges this gap by first retrieving candidates via fuzzy SAX matching and subsequently employing Python verification to rigorously enforce the constraint, ensuring precise grounding where pure model-based approaches fail.

## 7. Conclusion

In this paper, we formally define the **NLQ4TSDB** task. To facilitate evaluation, we introduced **NLQTSBench**, a hierarchical benchmark that necessitates reasoning over long histories. To address this challenge, we proposed **Sonar-TS**, a neuro-symbolic framework implementing a "Search-Then-Verify" pipeline. By orchestrating coarse-grained symbolic search and fine-grained algorithmic verification, the system effectively bridges the gap between abstract user intents and raw numerical data. Our experiments demonstrate that Sonar-TS successfully handles complex temporal reasoning tasks where traditional paradigms fail. This work serves as a foundational step toward DB-grounded time series intelligence, offering a robust baseline for future research in semantic indexing and intelligent data monitoring. See Appendix D for a discussion of limitations and future work.

## Acknowledgments

S. Pan was partially supported by the Australian Research Council (ARC) under grants FT210100097 and DP240101547, and the CSIRO – National Science Foundation (US) AI Research Collaboration Program. This work was also supported by the NVIDIA Academic Grant in Higher Education and Developer program. Z. Tan acknowledges financial support from the China Scholarship Council (CSC) and the hospitality of Griffith University, where part of this work was carried out during his visiting period. X. Liu was supported by the National Natural Science Foundation of China (Grant Nos. 62462034, 62562033, 62272205, 62272206) and the Natural Science Foundation of Jiangxi Province (Grant Nos. 20232ACB202008, 20242BAB25119).

## Impact Statement

This paper presents a framework for natural language querying of time series databases. The primary positive impact is lowering the technical barrier for analysts and operators to retrieve events, intervals, and summaries from TSDBs, which may improve decision-making in domains such as monitoring and operations.

However, deploying such systems introduces potential risks. Privacy and confidentiality concerns may arise if the model is applied to sensitive industrial data. While we do not foresee immediate negative societal consequences, we encourage practitioners to maintain strict access controls and validation protocols when applying this technology to sensitive or critical environments.

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

# A. Benchmark Details

This appendix provides the implementation details of NLQTSBench. We document the query templates used to instantiate concrete questions (Appendix A.1), the construction pipeline that produces ground-truth-annotated samples from these templates (Appendix A.2), and the evaluation metrics used to score solver outputs (Appendix A.3).

## A.1. Query Templates

Every query in NLQTSBench is instantiated from a parameterized template. Each template contains two kinds of slots: *structural* slots (e.g., {channel}, {time_window}), which are filled from dataset metadata, and *semantic* slots (e.g., {pattern_name}, {superlative}), which are drawn from a controlled vocabulary. Semantic slots are constrained so that only meaningful combinations are produced (e.g., steepest is paired only with spike or valley, never with plateau). We list all templates below, grouped by the four levels of the taxonomy.

**Level 1: Basic Operations**   Level 1 templates produce queries that map directly to deterministic numerical operations. We distinguish atomic retrieval (a single statistic or extremum) from rolling-window analysis (best window over a moving context).

---

### Level 1: Basic Operations Templates

**Type 1: Atomic Retrieval**

- **Global Aggregation**
  **Template:** What is the **{agg}** value of channel **{channel}** in **{time}**?
  **Args:** **{agg}** ∈ ["maximum", "minimum", "average", "median", "range"]
- **Temporal Localization**
  **Template:** At what exact timestamp did channel **{channel}** **{action}** in **{time}**?
  **Args:** **{action}** ∈ ["reach its maximum value", "first rise above **{threshold}**", . . . ]
- **Interval Discovery**
  **Template:** Find the longest period where channel **{channel}** remained above **{threshold}** in **{time}**.
  **Args:** **{threshold}** → $P_{80}$ (Dynamic 80th percentile of data)

- - - - - - - - - - - - - - - - - - - - - - - - - - - - - - - - - - - - - - - - - - - - - -

**Type 2: Rolling Window Analysis**

- **Sliding Window Statistics**
  **Template:** Which **{window_desc}** in **{time}** had the **{metric}** for channel **{channel}**?
  **Args:**  **{metric}** ∈ ["highest/lowest average", "highest variance", "largest range"];
      **{window_desc}** → Random Integer [3D, 60D]

---

**Level 2: Pattern Recognition**   Level 2 templates target morphological patterns. Each template specifies a target shape (e.g., plateau, spike) and a qualifier (longest, deepest, etc.) that selects a specific instance from the candidate set.

---

### Level 2: Pattern Recognition Templates

- **Shape Identification**
  **Template:** Identify the time range of the **{superlative}** **{pattern_name}** in channel **{channel}** within **{time_window}**.
  **Args:** **{pattern_name}** ∈ ["plateau", "upward spike", "deep valley", "step ascent/descent"];
      **{superlative}** ∈ ["longest", "highest", "deepest", "largest"]

- - - - - - - - - - - - - - - - - - - - - - - - - - - - - - - - - - - - - - - - - - - - - -

- **Periodicity Detection**
  **Template:** What is the dominant cycle period (in data points) of channel **{channel}** within **{time_window}**?
  **Args:** Target Signal ∈ ["sine", "cosine", "composite"];
      Period Range → Random Integer [30, 120] points

---

- **Subsequence Matching**
  **Template:** Analyze the reference pattern in **{query_window}**. Find the time interval where channel **{channel}** exhibits the most similar pattern within the search context **{search_window}**.
  **Args:** Injected Prototypes ∈ ["bell curve", "step pattern", "double-peak (M-shape)", "sharp spike"]

**Level 3: Semantic Reasoning**  Level 3 templates extend pattern recognition to longer horizons (typically year-scale) and to cross-channel reasoning. The system must contextualize events against global baselines or infer relational discrepancies between correlated series.

---

### Level 3: Semantic Reasoning Templates

- **Composite Trend Search (Top-K)**
  **Template:** Identify the top-**{k}** dates in channel **{channel}** during **{year}** that exhibit the most significant **{pattern_desc}** trend.
  **Args:** **{pattern_desc}** ∈ ["rapid rise then fall", "gradual reversal", "step ascent", ...];
  Injection Intensity → Linear Decay (ensures strict ranking)

- **Contextual Anomaly Detection**
  **Template:** Identify the period in channel **{channel}** during **{year}** that experienced the most significant **{anomaly_desc}**.
  **Args:** **{anomaly_desc}** ∈ ["severe flood" ($> 3\sigma$ surge), "severe drought" (variance $\to 0$)];
  Context → Full Calendar Year (>30k points)

- **Causal Anomaly Detection**
  **Template:** Given that channel **{upstream}** causes **{downstream}**, identify the time period in **{year}** where **{downstream}** shows a significant causal anomaly, such as an **{break_desc}**.
  **Args:** **{break_desc}** ∈ ["inverse trend against source", "flat line during activity"];
  Mechanism → Correlation Break Injection

---

**Level 4: Insight Synthesis**  Level 4 contains a single template that asks for a structured narrative report. To make the free-form output deterministically parseable, the prompt enforces a controlled vocabulary for trend descriptions and a fixed two-part schema.

### Level 4: Insight Synthesis Templates

- **Trend & Anomaly Audit (Report Generation)**
  **Template:** Analyze channel **{channel}** for the period **{target_month}**.
  **Constraint:** Use ONLY standardized phrases: ["rapid/gradual rise", "rapid/gradual fall", "steady/fluctuating stable"].
  **Requirement:** Provide a structured report covering: 1. Trend Segmentation (with precise HH:MM:SS timestamps). 2. Significant Outlier Audit (ignoring minor noise).
  **Args:** Scenario Generation → Chained injection of 2–4 distinct primitives (e.g., Linear Trend → Stable → Oscillation);
  **Ground Truth** → Structured Fact Sheet defining the exact start/end and type of every stage.

---

### A.2. Construction Pipeline

This appendix expands the three-stage pipeline summarized in Section 4.2. Background series are sampled from three real-world domains: CausalRivers (Stein et al., 2025), ETTm1 (Zhou et al., 2021), and SMD (Su et al., 2019).

**Stage 1: Parameterized Template Formulation.**  We design a small set of parameterized templates that cover common query intents, with the full template list given in Appendix A.1. Each template combines two kinds of slots: structural slots filled from dataset metadata, and semantic slots filled from a controlled vocabulary.

Structural slots (e.g., {channel}, {time_window}) are populated from the dataset's metadata. Semantic slots (e.g., {superlative}, {pattern_name}) are drawn from a predefined vocabulary under semantic constraints that ensure

logical coherence (e.g., `steepest` pairs only with `spike` or `valley`, never with `plateau`). This controlled instantiation expands a small number of base templates into a diverse query set while avoiding nonsensical combinations.

**Stage 2: Signal Injection.** This stage turns an instantiated template into a concrete time series sample. The injection is built from a library of parametric primitives, composed and then planted into a real background window with carefully calibrated amplitude. We elaborate the three steps below.

**(1) Function lifting.** Each morphological descriptor is mapped to a parametric base function $f(t)$ drawn from a primitive library that currently supports three families:

- *Transient patterns*, modeled by Gaussian kernels for localized events such as spikes or sharp valleys:

$$f_{spike}(t) = A \cdot \exp\left(-\lambda(t - t_0)^2\right), \tag{4}$$

  where $A$ controls the peak height, $\lambda$ controls the sharpness, and $t_0$ is the location.

- *State shifts*, modeled by dual-sigmoid activations for plateaus and step-like regimes:

$$f_{box}(t) = \sigma(k(t - t_s)) - \sigma(k(t - t_e)), \tag{5}$$

  where $\sigma(\cdot)$ is the sigmoid function, $[t_s, t_e]$ defines the support of the plateau, and $k$ controls the steepness of the rising and falling edges.

- *Oscillations*, modeled by composite sinusoids with Gaussian noise for periodic patterns:

$$f_{wave}(t) = \sum_{i=1}^{K} A_i \sin(\omega_i t + \phi_i) + \epsilon(t). \tag{6}$$

Because the primitives are differentiable and parametric, complex scenarios are built by composition. Level 2 uses a single primitive instance to test local pattern recognition. Level 3 either chains two primitives ($f_{rise} \oplus f_{fall}$) to form a composite trend, or breaks the correlation between paired channels for causal anomaly detection. Level 4 composes a multi-stage sequence of 2–4 primitives with optional distractors, simulating a full-lifecycle scenario for narrative reporting.

**(2) Background selection.** We retrieve a real-world background window $X_{bg}(t)$ from the metadata pool, matching the sampled $\{$`time_window`$\}$. The window then undergoes a stability check based on its local variance $\sigma^2(X_{bg})$: if the variance exceeds an upper bound, native fluctuations are of comparable magnitude to the injection and would obscure the planted pattern. Such windows are rejected, and the pipeline resamples another candidate. This filtering step ensures that the SNR calibration in Step (3) operates on a sensible base.

**(3) Superimposition with SNR calibration.** The synthetic sequence is generated by adding the transformed primitive onto the background:

$$X_{syn}(t) = X_{bg}(t) + \alpha \cdot \mathcal{T}(f(t)), \tag{7}$$

where $\mathcal{T}$ denotes temporal transformations such as shifting, scaling, and stretching, and $\alpha$ is an adaptive gain factor.

The key challenge is choosing $\alpha$. A fixed value would be invisible in a highly volatile window and unrealistically large in a stable one. We therefore define the signal-to-noise ratio (SNR) of a candidate injection as the ratio between the energy of the injected signal and the local variance of the background:

$$\text{SNR}(\alpha) = \frac{\mathbb{E}\left[(\alpha \cdot \mathcal{T}(f(t)))^2\right]}{\sigma^2(X_{bg})}, \tag{8}$$

and solve for $\alpha$ so that the resulting SNR falls inside a target band ($1.0 < \text{SNR} < 1.5$). Because the denominator is the local variance of $X_{bg}$, this forces $\alpha$ to scale with the native noise level, keeping the injected pattern simultaneously salient and distributionally consistent with the surrounding signal.

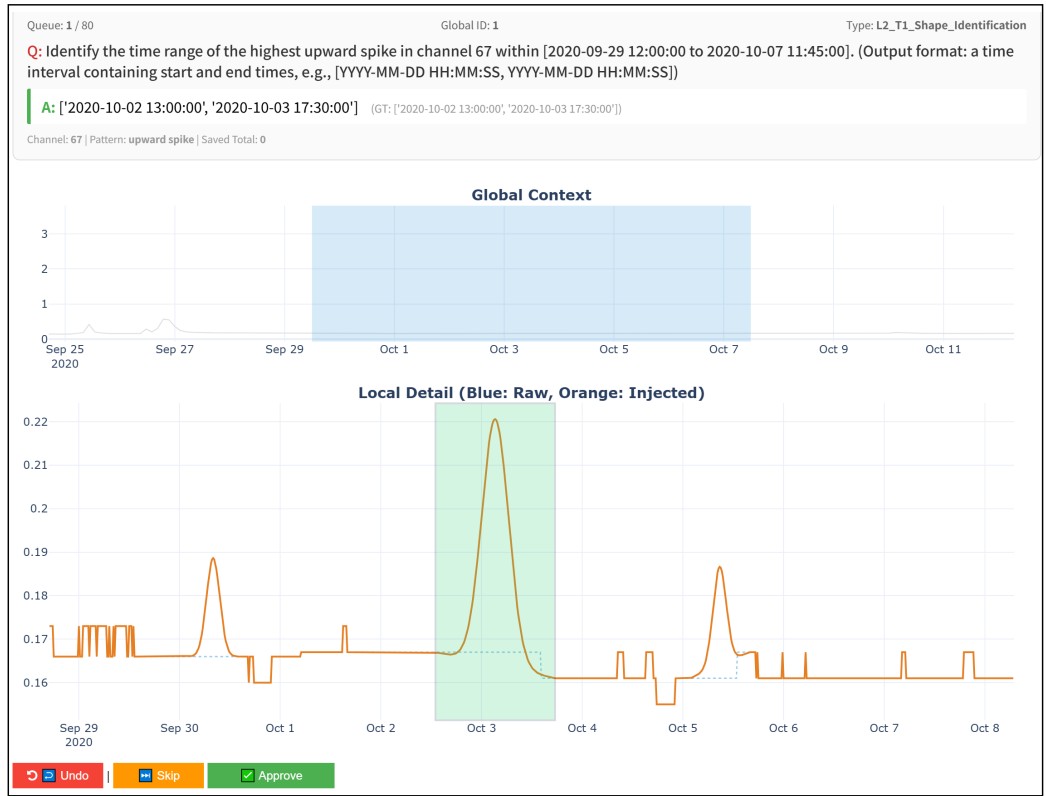

*Figure 6.* The human verification interface. Annotators inspect both the global context (top) and the local detail view (bottom, where the injected signal in orange is overlaid on the raw data in blue) to validate the ground truth.

**Stage 3: Human Visual Verification.** Automated checks alone cannot guarantee that a generated query is perceptually solvable. We therefore add a human-in-the-loop audit as the final safeguard. Expert annotators inspect rendered charts for each candidate sample through the interface shown in Figure 6: the global view confirms the injection fits its surrounding regime, while the local view confirms perceptual alignment with the linguistic query. Only samples that clear both the SNR check and human inspection enter the final benchmark.

### A.3. Evaluation Methods

NLQ4TSDB outputs span five formats: scalars, timestamps, intervals, date sets, and structured reports. We pair each format with a tailored metric. Table 6 provides a task-to-metric lookup at the end of this section.

**1. Scalars and Timestamps.** For **scalars** (e.g., a periodicity value), exact equality is brittle under floating-point arithmetic. We instead use a relative-accuracy score normalized to the magnitude of the ground truth:

$$\text{Score}_{\text{scalar}}(y_p, y_g) = \max\left(0,\ 1 - \frac{|y_p - y_g|}{|y_g| + \epsilon}\right), \tag{9}$$

where $\epsilon = 10^{-9}$ avoids division by zero. For **timestamps**, we use a binary hit rate: a prediction is correct if $|t_{pred} - t_{gt}| \le \delta$, where $\delta$ is the sampling resolution (typically 0, i.e., exact match).

**2. Time Intervals.** For tasks that return a continuous interval (e.g., Sliding Window, Shape Identification, Contextual Anomaly, Causal Anomaly), we evaluate the temporal overlap between the predicted interval $I_p$ and the ground truth $I_g$ via Intersection over Union:

$$\text{Score}_{\text{interval}}(I_p, I_g) = \frac{\text{duration}(I_p \cap I_g)}{\text{duration}(I_p \cup I_g)}, \tag{10}$$

where duration$(\cdot)$ is measured in seconds. Disjoint intervals score 0.

**3. Timestamp Sets.** For tasks that return an unordered set of dates (e.g., Top-K composite trends), we treat the prediction $D_p$ and ground truth $D_g$ as sets and compute F1:

$$\text{Score}_{\text{set}}(D_p, D_g) = 2 \cdot \frac{\text{Precision} \cdot \text{Recall}}{\text{Precision} + \text{Recall}}, \tag{11}$$

where Precision is the fraction of correct dates in the prediction, and Recall is the fraction of ground-truth dates successfully retrieved.

**4. Structured Reports.** Insight Synthesis (Level 4) returns free-form text. Surface metrics like BLEU and ROUGE cannot judge factual correctness in this setting, so we cannot rely on them. Because our prompt enforces a controlled vocabulary and a strict two-part schema (see Appendix A.1), the output can be deterministically parsed into Trend Segments and Outliers, which are then scored by a composite:

$$\text{Score}_{\text{report}} = 0.4\,S_{trend} + 0.3\,S_{interval} + 0.2\,S_{adj} + 0.1\,S_{outlier}, \tag{12}$$

The four sub-scores measure, respectively: (1) $S_{trend}$: alignment of trend types (rise vs. fall) via longest common subsequence. (2) $S_{interval}$: mean IoU over the matched trend segments. (3) $S_{adj}$: match rate of descriptive adjectives (rapid vs. gradual). (4) $S_{outlier}$: the detected anomaly timestamps.

The weights were calibrated to prioritize correct trend structure over fine-grained adjective matching.

*Table 6.* Task-to-metric lookup for NLQTSBench.

| Level | Subtask | Metric |
|---|---|---|
| L1 | Atomic Retrieval (Global Aggregation) | Relative Accuracy |
| L1 | Atomic Retrieval (Temporal Localization) | Hit Rate |
| L1 | Atomic Retrieval (Interval Discovery) | IoU |
| L1 | Sliding Window | IoU |
| L2 | Shape Identification | IoU |
| L2 | Periodicity Detection | Relative Accuracy |
| L2 | Subsequence Matching | IoU |
| L3 | Composite Trend | Set F1 |
| L3 | Contextual Anomaly | IoU |
| L3 | Causal Anomaly | IoU |
| L4 | Insight Synthesis | Composite Report Score |

**A.4. NLQTSBench-Lite Statistics**

We provide the per-task composition of NLQTSBench-Lite in Table 7. The Lite variant is purpose-built as a fixed evaluation suite for context-limited time series baselines (e.g., ChatTS, ITFormer, Time-R1) whose input window cannot accommodate the multi-year histories of the full benchmark.

*Table 7.* Task composition of NLQTSBench-Lite (500 instances in total).

| Level | Subtask | #Instances |
|---|---|---|
| L2 | Shape Identification (SI) | 100 |
| L3 | Composite Trend (CT) | 200 |
| L3 | Causal Anomaly (CsA) | 200 |
| **Total** | | **500** |

The three retained tasks correspond to the three core capabilities that NLQ4TSDB demands: *Shape Identification* for morphological grounding, *Composite Trend* for multi-stage temporal reasoning, and *Causal Anomaly* for cross-channel relational reasoning. Together they form a compact yet representative testbed for evaluating short-context models on the central challenges of the task.

# B. Implementation Details of Sonar-TS

## B.1. System Configurations

It is crucial to note that Sonar-TS is designed as a highly configurable framework. The hyperparameters described below represent our default instantiation for the benchmark evaluation. All configurations can be flexibly adapted to specific downstream applications or industrial deployments.

**Databases and Backbone Models.** The framework fundamentally decouples data storage from analytical reasoning. For the underlying TSDB, our current implementation targets SQL-compatible systems (e.g., InfluxDB). For the backbone LLM, our primary experiments utilize DeepSeek-V3 for both task planning and code generation. To demonstrate consistent efficacy across varying model capacities, we additionally evaluate the framework using the open-weights Qwen family.

**Multi-scale Feature Tables.** To facilitate efficient multi-granular querying, we materialize feature tables aligned with natural temporal hierarchies: `Yearly`, `Monthly`, and `Daily` views. Each row in these tables summarizes a specific time window $w = [t_{start}, t_{end})$ for a unique time series channel. The schema includes:

- **Metadata:** `series_id`, `view_type` (e.g., 'daily'), `window_start`, `window_end`.
- **Statistical Primitives:** `min_val`, `max_val`, `avg_val`, `std_val`, and `slope`.
- **Morphological Tokens:** `SAX` and `SAX_len`.

**Statistical Primitives** For each window, we precompute a set of lightweight descriptive statistics. While the current version of the NLQ4TSDB benchmark focuses heavily on complex pattern recognition, making simple statistics less central to the primary evaluation tasks, these primitives remain critical for optimizing broader query types. For instance, a query such as *"Find the longest daily interval with a continuous upward trend"* can be accelerated significantly by filtering on precomputed `slope` values (e.g., `slope > 0`), thereby pruning the search space before accessing raw high-frequency data. Similarly, `std_val` serves as an efficient proxy for filtering volatile or stable periods.

**Morphological Tokens (SAX Implementation).** To support shape-based retrieval, we implement Symbolic Aggregate approXimation (SAX), transforming complex continuous shapes into discrete string signatures.

1. **Piecewise Aggregate Approximation (PAA).** We first reduce the dimensionality of the raw time series sequence $C = \{c_1, \ldots, c_n\}$ within a window into a vector of length $w$, denoted as $\bar{C} = \{\bar{c}_1, \ldots, \bar{c}_w\}$. The choice of $w$ adapts to the temporal granularity to preserve semantic interpretability:

   - **Yearly View:** $w = 12$, aligning with months.
   - **Monthly View:** $w$ is dynamic, equal to the number of days in that specific month (e.g., 28, 30, or 31).
   - **Daily View:** $w = 24$, aligning with hours.

   The $i$-th PAA coefficient is computed as the mean of the corresponding segment:

   $$\bar{c}_i = \frac{w}{n} \sum_{j=\frac{n}{w}(i-1)+1}^{\frac{n}{w}i} c_j \tag{13}$$

2. **Symbolic Mapping.** The PAA vector $\bar{C}$ is Z-normalized to have a mean of zero and a standard deviation of one. We then map each coefficient $\bar{c}_i$ to a symbol $s_i$ from an alphabet $\Sigma$ of size $\alpha$ (in our implementation, $\alpha = 5$, $\Sigma = \{'a', 'b', 'c', 'd', 'e'\}$). The mapping is defined by a set of breakpoints $\beta = \{\beta_0, \ldots, \beta_\alpha\}$, which divide the area under the Normal distribution $N(0,1)$ into $\alpha$ equiprobable regions. The mapping function is:

   $$s_i = \text{char}(j) \quad \text{if } \beta_{j-1} \leq \bar{c}_i < \beta_j \tag{14}$$

   where $\beta_0 = -\infty$ and $\beta_\alpha = \infty$. This results in a discrete signature string $\hat{C} = s_1 s_2 \ldots s_w$ that robustly encodes the shape of the time series window.

Figure 7 illustrates this SAX transformation process across different granularities.

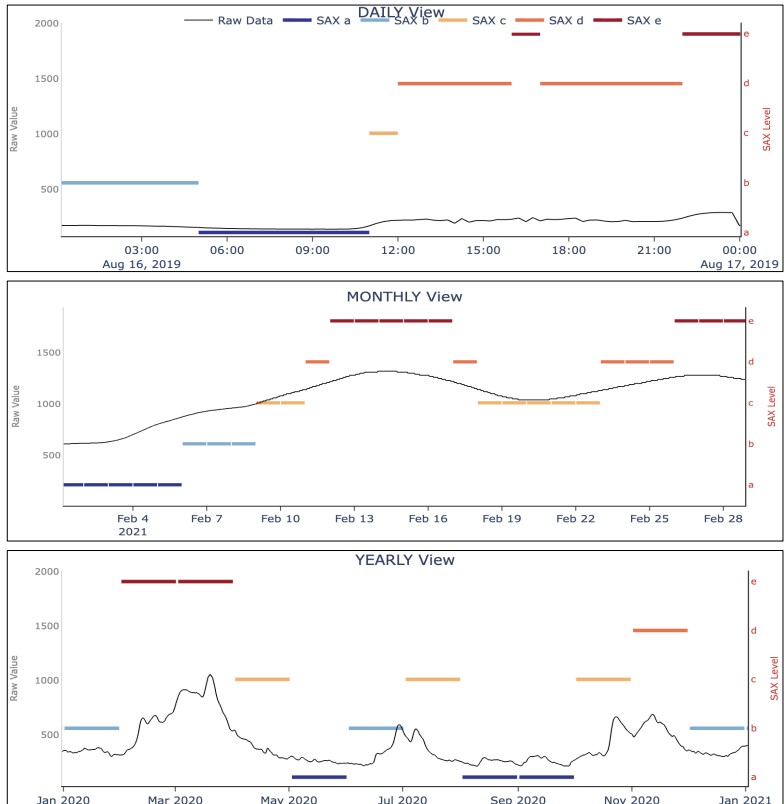

*Figure 7.* Visualization of Multi-Scale SAX Representations. The framework discretizes time series data across hierarchical granularities to support pattern matching at different resolutions: **(Top)** The *Daily View* captures high-frequency local fluctuations; **(Middle)** The *Monthly View* summarizes intermediate trends; **(Bottom)** The *Yearly View* abstracts long-term seasonality. The colored horizontal bars represent the assigned SAX symbols (from level $a$ to $e$) overlaid on the raw data (gray line).

**Experience Initialization (Prompt Cold Start).** To construct the initial Prompt Experiences without causing data leakage on the test set, we utilized a separate, hold-out dataset of 110 queries entirely independent of the evaluation benchmark. The "Reward-Summarize-Update" pipeline was executed on this set to bootstrap the initial knowledge base. Furthermore, the global Experience set is strictly capped at 20 items. This limit ensures that the entire set can be directly injected into the system prompt for every online query without risking context overflow or distracting the LLM.

**Runtime Self-Correction.** To enhance system robustness during the online "Execution & Self-Correction" phase, the system employs an automated debugging loop. We bound the maximum retry limit to $R = 3$. Empirical observations indicate that the majority of recoverable execution errors (e.g., empty SQL results or Python syntax faults) are resolved within this window, and further retries yield diminishing returns.

### B.2. Prompt Design

We implement a structured prompting pipeline to orchestrate the Search-Then-Verify workflow. This process dissociates high-level logical planning from low-level code implementation, featuring a dual-mode code generator capable of iterative self-correction.

**1. Task Planner Prompt.** The Planner is responsible for selecting the execution mode and the data source. As shown in the prompt below, the model is explicitly guided to distinguish between the "High IO" raw data table (Wide) and the "Low IO" feature tables (Long) to optimize retrieval efficiency.

## Task Planner Prompt

You are a Task Planner for Time Series Analysis. Decompose the question into a JSON plan. No code.

— **DATABASE SCHEMA** —
{database schema}

1. Raw Data (data):
High I/O cost. Use ONLY when the time range is explicitly known and precise values are needed.
2. Feature Tables (feature_*):
Low I/O cost. Contains precomputed stats (avg, std) and shape descriptors (sax). Use for searching patterns or scanning large historical ranges.

— **EXPERIENCES** —
{experiences}

— **User Question** —
{Input Question}

— **PLANNING STRATEGY** —
Analyze the query constraints to decide the Pipeline Mode:

Mode A: DIRECT_ACCESS (Fetch → Compute)
Criteria: 1. Time range is fixed (e.g., "last 24h", "in 2023"). 2. Task involves calculation on a specific segment.
Strategy: Direct SQL query on "data" → Python Processing.

Mode B: SEARCH_THEN_VERIFY (Prune → Verify)
Criteria: 1. Time range is UNKNOWN/OPEN (e.g., "Find the period where..."). 2. Logic is MORPHOLOGICAL (e.g., "V-shape") or requires historical context.
Strategy: Search "feature_*" filters → Get Candidates → Python Verification.

— **Output Format (Strict JSON)** —
```
{
    "reasoning": "Analysis of time range constraints and search intent",
    "pipeline_mode": "DIRECT_ACCESS" or "SEARCH_THEN_VERIFY",
    "step_1_retrieval": {
        "target_table": "data" or "feature_daily" or "feature_monthly",
        "sql_logic_hint": "Description of the SQL filter/logic"
    },
    "step_2_computation": {
        "needs_python": true,
        "logic_description": "Description of the math/verification logic"
    }
}
```

**2. Code Generator Prompt.** Following the plan, the Code Generator constructs the executable queries. It operates in two modes: Generation (synthesizing the initial code) and Refinement (fixing errors based on runtime feedback). In the generation phase, strict environmental constraints and a domain-specific operator library are enforced to ground the model's logic. Subsequently, the refinement phase acts as a runtime debugger, allowing the agent to iteratively self-correct against dynamic exceptions (e.g., empty SQL results) that are unpredictable during the initial generation.

## Code Generation Prompt (Generation Mode)

You are the Code Generator. Translate the structured Task Plan into executable Python code.

— **DATABASE SCHEMA** —
{database schema}

— **EXPERIENCES** —
{experiences}

**— User Question —**
{Input Question}

**— TASK PLAN (JSON) —**
{JSON Plan generated by Task Planner}

**— CODE GENERATION STRATEGY —**
Generate a single Python code block to execute the plan.

1. Execution Environment:
You have access to pandas (pd), numpy (np), and scipy.
The variable "conn" is already defined and connected to the database.
- DO NOT create a new connection. DO NOT close "conn".
- Execute SQL using: df = pd.read_sql_query(sql, conn).

2. Operator Library (Module: sonar_ops):
Prioritize using the following pre-defined atomic operators for verification.
If a task cannot be solved by these operators, synthesize standard pandas/numpy logic.
- detect_period(data, max_lag): Estimates the dominant cycle length.
- find_best_match(query, search, metric): Finds the most similar subsequence via DTW.
- detect_changepoints(data, penalty): Identifies structural break points (PELT).
- calc_trend_slope(data): Computes robust slope (Theil-Sen) for trend description.
- calc_correlation(seq_a, seq_b, lag): Measures statistical relationship between series.

**— Output Format —**
Return ONLY the executable Python code block. No markdown explanation.
The code must define a final variable final_answer containing the result.

## Code Generation Prompt (Refinement Mode)

You are the Code Refinement Agent. The previous execution failed. Analyze the error trace and modify the code to make it executable.

**— SHARED CONTEXT —**
*(Includes Database Schema, Experiences, Input Question, Task Plan, and Code Generation Strategy)*

**— HISTORY: TURN 1 —**
Code:
{Previous Python Code}

Execution Feedback:
{Error Traceback OR "Empty Result"}

**... (History stacks up to 3 turns) ...**

**— Output Format —**
Return ONLY the corrected executable Python code block. No markdown explanation.
The code must define a final variable final_answer containing the result.

**3. Experience Summarizer Prompt.** This module acts as a "Technical Lead" conducting a post-mortem. It analyzes the full execution trajectory—including the initial plan, the challenges faced (error history), and the final working code—to extract a single, high-value insight. This ensures that the system learns not just from success, but from the corrections applied during the process.

## Experience Summarization Prompt

You are a Technical Lead conducting a Post-Mortem. Summarize the technical solution.

**— Question —**
{Input Question}

— **Plan** —
{JSON Plan generated by Task Planner}

— **CODE & EXECUTION HISTORY** —
{Full Trace of Code Generations and Execution Feedbacks}

— **Task** —
Extract ONE concise, reusable technical insight.
- How was the task resolved?
- Analyze the reasons for success or failure.
- Generalize the finding.

— **Output Format** —
Insight: [Your 1-sentence summary]

**4. Experience Updater prompt**   To prevent the experience list from growing indefinitely, the Updater functions as a "Knowledge Curator." It takes the newly extracted insight and merges it into the existing Global Experience Pool. The model is instructed to perform semantic operations—**Add** (if new), **Merge** (if similar), or **Discard** (if redundant)—ensuring the knowledge base remains compact and highly relevant.

---

**Experience Updater Prompt**

You are the Knowledge Base Curator for Sonar-TS. Manage the global experience list.

— **EXISTING EXPERIENCES** —
{Current List of N Insights}

— **NEW INSIGHT** —
{Output from Summarizer}

— **UPDATE STRATEGY** —
Compare the New Insight with the Existing List and apply one operation:
1. **ADD**: If the insight covers a new edge case.
2. **MERGE**: If a similar insight exists, combine them into a more robust rule.
3. **DISCARD**: If the insight is trivial or fully covered.
Constraint: Keep the total list size under 20 items to preserve context window.

— **Output Format** —
Return the updated list of experiences (JSON list of strings).

---

## C. Additional Experiments

We present three additional analyses that probe Sonar-TS from different angles: (1) sensitivity to two key parameters, namely the LLM backbone and the SAX alphabet size (Appendix C.1); (2) the internal effectiveness of the Search stage that anchors our Search-Then-Verify workflow (Appendix C.2); and (3) computational overhead relative to a comparable baseline (Appendix C.3).

### C.1. Parameter Sensitivity

We examine the sensitivity of Sonar-TS to two key parameters: the choice of LLM backbone and the SAX alphabet size $\alpha$.

**Sensitivity to LLM Choice.**   Table 8 reports the performance of Sonar-TS across underlying LLMs of varying scales. As expected, scaling up the model size consistently yields better overall performance, with our default DeepSeek-V3 achieving the highest average score. While the Experiences memory mitigates the zero-shot reasoning burden by providing historical analytical skills, the unconventional "SAX + regex" querying paradigm still demands strong capabilities from the underlying LLM: the backbone must translate subjective morphological descriptions into precise SAX-based regular expressions. Smaller models (e.g., Qwen3.5-9B) therefore struggle with this morphological-to-symbolic mapping, whereas larger models handle the translation reliably.

*Table 8.* Sensitivity of Sonar-TS to the LLM backbone.

| Backbone Model | Avg. Score |
|---|---|
| Qwen3.5-9B | 0.3598 |
| Qwen3.5-27B | 0.4817 |
| Qwen3.5-35B | 0.5359 |
| DeepSeek-V3 (Default) | **0.6144** |

**Sensitivity to SAX Alphabet Size.** Table 9 evaluates the effect of varying the SAX alphabet size $\alpha$ on morphology tasks (SI, CT). The optimal value is $\alpha = 5$, driven by two factors. First, *semantic alignment*: natural language typically uses five morphological gradations (e.g., from "very low" to "very high"), so setting $\alpha = 5$ maps these descriptors directly onto symbolic tokens. Second, *resolution trade-off*: a larger alphabet ($\alpha = 7$) fractures the search space and complicates robust regex generation, while a smaller alphabet ($\alpha = 3$) lacks the expressive power to capture necessary morphological nuances.

*Table 9.* Sensitivity of morphology tasks to the SAX alphabet size $\alpha$.

| Alphabet Size ($\alpha$) | L2: SI | L3: CT |
|---|---|---|
| $\alpha = 3$ | 0.1927 | 0.2318 |
| $\alpha = 5$ (Default) | **0.3336** | **0.2988** |
| $\alpha = 7$ | 0.2896 | 0.2675 |

## C.2. Search Effectiveness

A core challenge of NLQ4TSDB is morphological reasoning and retrieval, which Sonar-TS addresses via the Search-Then-Verify workflow. To examine the internal effectiveness of this mechanism, we focus on the morphology-dependent tasks: Shape Identification and Composite Trend. Table 10 reports the intermediate retrieval performance of the initial Search stage on these tasks.

The Search stage narrows the search space by orders of magnitude relative to the raw time series (typically containing tens of thousands of points), while maintaining moderate recall. This validates our core hypothesis: symbolic representation enables morphological retrieval. We acknowledge that the current recall indicates room for improvement, which we attribute to two factors. First, the task is inherently hard: translating subjective morphological concepts into strict matching conditions is highly prone to semantic ambiguity. Second, our SAX-based symbolic implementation suffers inevitable information loss during compression, which hinders fine-grained morphological retrieval; we discuss this limitation further in Appendix D. As an initial attempt, these results nonetheless demonstrate the feasibility of our retrieval strategy. Exact candidate counts and recall values may vary with benchmark configuration and future SAX index improvements; we refer readers to our open-source repository for the most up-to-date numbers.

*Table 10.* Internal evaluation of the Search stage, reporting average candidates retrieved and recall. Absolute candidate counts depend on benchmark construction details (e.g., the number of injected positives per query and the temporal scope of each instance).

| Task Category | Avg. Candidates | Recall |
|---|---|---|
| L2: Shape Identification | 9.3 | 0.5482 |
| L3: Composite Trend | 5.8 | 0.4713 |

## C.3. System Overhead

Table 11 reports the system overhead of Sonar-TS along four practical metrics: offline construction time for the feature tables, online querying latency, average token consumption per query, and relative storage overhead. We use MAC-SQL as the baseline because its multi-agent collaborative design closely aligns with Sonar-TS, ensuring a fair and architecturally comparable evaluation.

Sonar-TS exhibits an online querying latency of roughly 22 seconds and consumes approximately 11k tokens per query. While slightly higher than the Text-to-SQL baseline MAC-SQL, this overhead represents a reasonable trade-off given the

*Table 11.* System overhead and computational cost of Sonar-TS. Time is in seconds (s), token cost is the per-query average, and storage overhead is relative to the raw data size. Reported figures reflect a representative deployment on our experimental setup; concrete values depend on hardware, API latency, and code version.

| Efficiency Metric | MAC-SQL | Sonar-TS |
|---|---|---|
| Offline Construction Time | N/A | ∼83 s |
| Online Querying Latency | ∼15 s | ∼22 s |
| Average Token Cost | ∼8k | ∼11k |
| Storage Overhead | N/A | 11% |

complexity of NLQ4TSDB. Conceptually, Sonar-TS functions as a superset of Text-to-SQL methods: it retains foundational SQL generation capabilities but additionally orchestrates multi-step task planning, morphological retrieval, and Python code execution. Given these supplementary operations, the modest latency increase remains within an acceptable range for the target use case. As our system continues to evolve, latency and token cost may shift with code optimizations and changes in upstream LLM inference speed; we refer readers to our open-source repository for current framework.

## D. Limitations and Future Work

While Sonar-TS demonstrates the feasibility of the NLQ4TSDB task, several limitations remain.

**Realism of Data and Queries.** Due to the nascent nature of the NLQ4TSDB task, there is a severe scarcity of available datasets. We initially sought to use real-world data but faced significant bottlenecks: manually annotating million-point sequences is prohibitively labor-intensive, and reliable automated labeling algorithms are lacking. Consequently, we relied on a synthetic-injection pipeline (Section 4.2) to guarantee verifiable ground truths. While this ensures an initial testbed, aligning with the true distribution of real-world data and authentic user queries remains a critical necessity. Future work must focus on developing realistic datasets aligned with actual business demands.

**Information Loss in Symbolic Representation.** To enable morphological retrieval over continuous signals, we employ SAX. However, its reliance on rigid temporal and amplitude bins inevitably induces information loss, causing the system to inherently struggle with fine-grained local fluctuations. Future iterations should explore learnable discrete representations. Transitioning from these rigid heuristics to semantics-aware tokenizers could provide the nuanced, high-fidelity representations specifically tailored for complex NLQ4TSDB tasks.

**Scalability of Experience Injection.** Because general LLMs lack the specialized domain knowledge for complex time series querying, we implement a *prompt cold start* process (Section 5.3) to extract essential analytical skills as *Experiences*. Currently, to ensure execution stability, we inject a hard limit of 20 items directly into the system prompt. While this pragmatic workaround is effective, it is not a scalable long-term solution. As user intents diversify and databases expand, a static set of skills will inevitably become insufficient. Future work should develop dynamic retrieval modules to scale for increasingly complex analytical scenarios.

