# OpenReview forum: "Sonar-TS: Search-Then-Verify Natural Language Querying for Time Series Databases"
_ICML.cc/2026/Conference — ICML 2026 regular_

### Official Review · Reviewer_yUKc · 2026-03-02

**Soundness:** 3
**Presentation:** 3
**Significance:** 2
**Originality:** 2
**Overall Recommendation:** 4
**Confidence:** 2

**Summary:**

This paper proposes Sonar-TS, which is a neuro-symbolic framework that tackles NLQ4TSDB via a so-called Search-Then-Verify pipeline. It uses a feature index to ping candidate windows via SQL, followed by generate Python programs to lock on and verify candidates against raw signals. They also propose NLQTS-Bench, which is the large-scale benchmark designed for NLQ over TSDB-scale histories. Experiments demonstrate good performance.

**Compliance With Llm Reviewing Policy:**

Affirmed.

**Final Justification:**

The paper is well-written and the idea is clear. The proposed model is reasonable, and the experimental results are overall satisfactory.

**Key Questions For Authors:**

1. Could the authors provide a detailed explanation for why the proposed model achieves lower scores on simple SI tasks compared to CT tasks? Additionally, is the SAX compression component indispensable, or could it be replaced with alternative representations? If so, are there potentially better schemes that might improve performance or efficiency?
2. Is the proposed model sensitive to the choice of the underlying LLM? An ablation study across LLMs of varying scales is needed to assess how model size impacts performance. Additionally, an analysis of time efficiency is also necessary to evaluate the computational cost and practical feasibility of the approach.
3. I understand the scarcity of appropriate datasets in this domain. However, validating the model on a broader range of datasets would better demonstrate its generalizability and applicability. Would it be possible to include additional datasets for evaluation, rather than validating solely on the proposed NLQTSBench? If possible, please provide corresponding experimental results.

**Limitations:**

No, there is no discussion about the limitations of the proposed model.

**Strengths And Weaknesses:**

Strengths:
1. This paper is well-written and presents a clear, focused idea. The exposition is systematic and easy to follow, with the problem background and model design being naturally motivated.
2. The search-then-verify workflow which includes task planning, code generation, and execution and self-correction is clear and reasonable.
3. It proposes the NLQTSBench which is a large-scale benchmark that fills a critical gap in the availability of such datasets for the community.

Weakness:
1. The related work section is somewhat weak and would benefit from additional details to help readers without extensive background in the area better contextualize the contributions. A more comprehensive discussion of prior methods and their limitations would improve clarity and accessibility.
2. I also have several concerns, please see the questions below.

---

> ### Author Rebuttal · Authors · 2026-03-31
>
> > **W 1:** The related work section lacks details on prior methods and limitations.
>
> We will expand the Related Work section in the revised paper.
>
> > **KQ 1:** Why are SI scores lower than CT scores? Can SAX be replaced?
>
> **1. SI vs. CT scores.**
>
> We address this observation from two perspectives:
>  * **Inherent task difficulty:** In Sonar-TS, Shape Identification (SI) is not inherently simpler than Composite Trend (CT). Both require the LLM to translate morphological intents into complex regular expressions, making their baseline reasoning difficulty quite similar.
>
> * **The impact of SAX compression:** SAX compression inherently penalizes sharp morphological shapes more than sustained trends. For an SI task, a sharp "V-shape valley" ideally mapping to `cbabc` might flatten into `cbbbc` under compression, causing a direct retrieval miss. Conversely, a CT task like "rise then stable" (`abcddd`) might compress into `abccdd`. Fundamentally, compression discards fine-grained local details. These precise details are highly critical for SI tasks, whereas CT tasks are much more resilient to this information loss as they focus on broader macroscopic transitions.
>
> **2. SAX alternatives:**
>
> Yes, SAX can be replaced. Sonar-TS only requires a time-series symbolization method. Adopting an alternative approach with lower information loss would directly enhance the system's overall performance.
>
> > **KQ 2:** Is the model sensitive to the choice of LLM? Could you analyze time efficiency?
>
> **1. LLM sensitivity:**
>
> We supplemented a sensitivity analysis across LLMs of varying scales. As shown below, larger models intuitively achieve better performance.
>
> **Table R4-1: LLM Sensitivity Analysis**
>
> | Backbone Model | Avg. Score |
> | :- | :- |
> | Qwen3.5-9B | 0.3598 |
> | Qwen3.5-27B | 0.4817 |
> | Qwen3.5-35B | 0.5359 |
> | DeepSeek-V3.2 (Default) | 0.6168 |
>
> While our "Experiences" memory reduces the reasoning burden by providing historical skills, the unconventional "SAX + regex" paradigm inherently demands robust capabilities from the LLM. The model must rely on abstract reasoning to successfully map  morphological patterns into SAX regular expressions.
>
> **2. Efficiency analysis:**
>
> Please refer to our response to **Reviewer e2f8 (KQ 4)** for details. To briefly summarize, Sonar-TS averages ~22.0 seconds per task for online querying.
>
> > **KQ 3:** I understand the scarcity of appropriate datasets in this domain. However, validating the model on a broader range of datasets would better demonstrate its generalizability and applicability.
>
> We sincerely appreciate your understanding. To demonstrate generalizability as requested, we evaluated Sonar-TS on established public benchmarks from two closely related fields: Text-to-SQL and TSQA.
>
> **Table R4-2: Proxy Evaluations on Related Domains**
>
> | System | Spider [1] (Text-to-SQL) | BIRD [2] (Text-to-SQL) | ChatTS [3] (TSQA) | Time-MQA [4] (TSQA) |
> | :--- | :--- | :--- | :--- | :--- |
> | Sonar-TS | 0.802 | 0.592 | 0.589 | 0.54 |
> | GPT-4o | 0.784 | 0.562 | 0.542 | 0.58 |
>
> *(Note: For ChatTS, we used Dataset A. For Time-MQA, we evaluated on the MCQ task within Open-Ended QA.)*
>
> As the results show, Sonar-TS maintains competitive performance across different domains. We objectively summarize the findings and implications of these proxy evaluations:
>
> * **Strong baseline reasoning:** Sonar-TS matches or exceeds strong baselines like GPT-4o on these proxy tasks. It successfully relies on its foundational agent framework and basic SQL generation pipeline to solve them.
>
> * **The limitation of proxy tasks:** However, these proxy datasets differ fundamentally from NLQ4TSDB. Standard Text-to-SQL lacks time-series data. Standard TSQA uses extremely short context windows. Therefore, Sonar-TS rarely activates its specialized morphological feature tables during these tests.
>
> * **The necessity of NLQTSBench:** Because proxy tasks cannot evaluate morphological retrieval, a dedicated benchmark is indispensable. NLQ4TSDB is a frontier research area. While NLQTSBench relies on synthetic data, its immediate goal is not to perfectly align with real-world complexities. Instead, it serves as a necessary testbed to verify whether the NLQ4TSDB task is fundamentally solvable. We believe it meets the research needs of this early stage.
>
> ---
>
> **Concluding Remarks:**
>
> We sincerely thank you for your feedback and for recognizing the value of our work. Your comments have greatly improved our paper. As suggested, we will expand the Related Work and add the requested supplementary details in the revised paper.
>
> We also appreciate your suggestions to evaluate the framework's generalizability. As foundational work in a brand-new research frontier, our primary goal is to bring the NLQ4TSDB problem to the community’s attention. We expect this initial exploration to draw more researchers to the field and drive forward progress on the NLQ4TSDB problem.
>
> Thank you again for your time, and we look forward to your final evaluation.

---

> > ### Author Rebuttal · Reviewer_yUKc · 2026-04-03
> >
> > Thanks for the detailed response. I have read the rebuttal and my concerns are basically addressed. Thus, I decide to keep the score.

---

> > > ### Author Response · Authors · 2026-04-07
> > >
> > > We sincerely thank you for your time and for confirming that our rebuttal has fully resolved your concerns. We deeply appreciate your constructive suggestions, which were incredibly valuable and have strengthened our paper.
> > >
> > > Thank you again for your support and encouragement of this foundational work in the NLQ4TSDB domain.

---

### Official Review · Reviewer_YMvU · 2026-03-11

**Soundness:** 3
**Presentation:** 3
**Significance:** 2
**Originality:** 2
**Overall Recommendation:** 4
**Confidence:** 4

**Summary:**

This paper studies natural language querying for time series databases, where user requests may involve temporal patterns, anomalies, intervals, and higher-level summaries over very long histories rather than standard relational lookups. The paper formalizes this setting as NLQ4TSDB and distinguishes it from both Text-to-SQL and time-series QA by its need for long-horizon evidence localization and morphological grounding. To address this, the paper proposes Sonar-TS, a neuro-symbolic Search-Then-Verify framework that first uses SQL over feature-indexed tables to retrieve candidate windows efficiently and then uses generated Python programs to verify the candidates on raw signals. The paper also introduces NLQTSBench, a benchmark for this setting with multiple difficulty levels and output types. The experiments suggest that Sonar-TS substantially outperforms prior alternatives on the proposed benchmark, and the ablations support the importance of feature-table search, prompting experience, and the verification stage.

**Compliance With Llm Reviewing Policy:**

Affirmed.

**Final Justification:**

The author addressed my concerns during the rebuttal process, and I have raised my score.

**Key Questions For Authors:**

see D1-D4

**Limitations:**

yes

**Strengths And Weaknesses:**

S1. The paper addresses an important and timely problem that is not well captured by standard Text-to-SQL or existing time-series QA benchmarks. Querying large time series databases with natural language is a meaningful setting with clear practical value.
S2. The main idea is well motivated. The Search-Then-Verify decomposition is intuitive and technically sensible for TSDB-scale data, since it combines efficient coarse retrieval with exact verification on raw signals.
S3. The benchmark contribution is valuable. The paper does more than present a system by also formalizing the task, proposing a taxonomy, and providing an evaluation suite with ablations that support the contribution of the main components.
D1. The main weakness is the realism of the benchmark. It is not fully clear how closely the benchmark queries reflect naturally occurring user intents and language, as opposed to controlled or injected constructions. This affects the external validity of the empirical results.
D2. The benchmark is built on a single real-world dataset, which limits the evidence for cross-domain realism and external validity. As a result, it is still unclear whether the proposed task formulation and query distribution are broadly representative of practical NLQ over time-series databases, rather than being tied to one particular domain.
D3. Some implementation choices appear fairly fixed and are not sufficiently stress-tested. For example, the paper uses a fixed SAX alphabet size and hierarchical segmentation settings (e.g., w = 24 for daily windows), but it is unclear how sensitive the system is to data granularity, noise level, or domain-specific temporal structure. This makes it difficult to judge how robust the approach would be beyond the reported setting.
D4. The paper includes a runtime self-correction loop, but the retry policy appears relatively coarse-grained. The current description suggests a fixed retry budget with execution feedback, without clearly distinguishing between different failure modes such as empty retrievals, syntax errors, and deeper semantic mistakes. As a result, it is unclear how effective the correction mechanism is in avoiding unproductive retries.

---

> ### Author Rebuttal · Authors · 2026-03-31
>
> > **D1:** The main weakness is the realism of the benchmark.
>
> NLQTSBench is currently built on templates and injected data. This approach is a necessary compromise for this nascent field:
>
> * **The ground truth challenge:** Localizing morphological segments across years of historical data makes manual annotation highly impractical. Injected data remains the most reliable method to establish precise ground truth for benchmarking. Using synthetic data is also a recognized practice in recent TSQA research [1].
>
> * **A "0-to-1" foundational step:** Because NLQ4TSDB is in its infancy, there is a scarcity of available data. Our immediate priority is establishing a reliable testbed to verify the core solvability of the NLQ4TSDB problem. Aligning with real-world queries requires massive resources. We must first validate the system's capabilities in a controlled environment before expanding to higher realism.
>
> * **Proxy evaluations:** For external validity, we evaluated Sonar-TS on TSQA and Text-to-SQL benchmarks. See our response to **Reviewer yUKc (KQ 3).**
>
> *[1] Xie, Z., et al. Chatts: Aligning time series with llms via synthetic data for enhanced understanding and reasoning. Proc. VLDB Endow., 18(8), 2025.*
>
> > **D2:** The benchmark is built on a single real-world dataset.
>
> We chose CausalRivers for the following reasons:
>
> * **Natural noise canvas:** Unlike pure synthetic data, which is often monotonous, CausalRivers provides natural noise and randomness. This increases data complexity.
>
> * **Domain-agnostic task design:** All tasks focus on numerical and geometric behaviors (e.g., longest plateau). These querys target underlying shapes instead of specific domain meanings.
>
> * **Infancy of the field:** While real-world alignment is our ultimate goal, our immediate objective is verifying the fundamental feasibility of solving NLQ4TSDB task. NLQTSBench serves as a necessary foundational testbed for this validation.
>
> > **D3:** The system appears bound to fixed parameters, and lacks stress testing across data granularities.
>
> Sonar-TS is designed as a flexible framework rather than a rigid set of parameters. We clarify our design philosophy below:
>
> * **Configurable parameters:** System variables are fully adjustable. The default setting of $w=24$ intuitively divides a daily cycle into 24 hourly segments. Users can easily modify this to match their specific data granularity.
>
> * **Rationale behind parameter choices:** Our settings directly align with the benchmark's task requirements. An alphabet size of 5 maps naturally to the five morphological trends we evaluate: rapid drop, slow drop, stable, slow rise, and rapid rise. This vocabulary can be expanded for finer trend resolution.
>
> * **The significance of the framework itself:** The primary contribution of Sonar-TS is the "Search-Then-Verify" paradigm rather than exhaustive parameter optimization. While we acknowledge the necessity for broader stress testing, conducting such evaluations requires diverse datasets that are currently scarce. As a foundational work in a nascent field, we hope to provide a robust baseline architecture for the community.
>
> > **D4:** The retry policy appears coarse-grained, making its effectiveness in avoiding unproductive retries unclear.
>
> We categorize execution feedback into two distinct types: execution errors and empty results, as detailed in **Appendix B.2.2.**
>
> For the detailed ablation data regarding the retry mechanism, please refer to our response to **Reviewer D3jW (KQ 2).** To briefly summarize the results, removing the self-correction module drops the system's average score from 0.6168 to 0.5913. This demonstrates that while the retry mechanism provides a performance improvement, its overall impact is relatively small.
>
> ---
>
> **Concluding Remarks:**
>
> We sincerely thank you for your detailed and constructive feedback. We appreciate that you find the NLQ4TSDB task highly meaningful. We also clearly hear your two main concerns regarding the realism of the benchmark and the robustness of our technical settings.
>
> Our perspective on this work is that NLQ4TSDB is frontier research. The field is previously unexplored and currently lacks foundational data. Therefore, our primary goal at this preliminary stage is to verify whether database-scale morphological retrieval is feasible. We are not aiming for immediate real-world deployment. We certainly do not claim to solve this complex problem perfectly in a single paper.
>
> Instead, we hope this work serves as a starting point. By proving the NLQ4TSDB task is tractable, we aim to attract the community's attention to this emerging problem and encourage more researchers to drive it forward. We are grateful for your time in reviewing our paper and look forward to your feedback.

---

> > ### Author Rebuttal · Reviewer_YMvU · 2026-04-06
> >
> > Thanks for the rebuttal. However, I will keep the score as the authors only provide the explantions for why choosing the benchmark and CausalRivers.

---

> > > ### Author Response · Authors · 2026-04-07
> > >
> > > We note your decision to maintain your score. However, for the public record, we wish to correct a factual inaccuracy in your final statement.
> > >
> > > **1. Empirical Evidence Provided**
> > > We provided new empirical evidence to directly address your concerns. Specifically, in our second-round response to Reviewer D3jW, we added:
> > >
> > > * **Cross-Domain Validity:** We evaluated Sonar-TS on a new energy dataset, ETTm1. The results are highly consistent and directly prove the framework's cross-domain generalizability.
> > > * **Implementation Choices:** We provided empirical ablation tables for the SAX alphabet sizes. The data proves that an alphabet size of 5 is the optimal choice.
> > >
> > > We encourage you to review these concrete experimental results.
> > >
> > > **2. Configurable Defaults vs. Algorithmic Limitations**
> > > We must also clarify your critiques about implementation details, such as `w=24` for daily windows and the retry limit.
> > >
> > > These are standard and fully adjustable engineering defaults. They are not algorithmic limitations. A proper evaluation of systems research should focus on the core methodology, rather than penalizing fully configurable parameters.

---

### Official Review · Reviewer_D3jW · 2026-03-13

**Soundness:** 2
**Presentation:** 3
**Significance:** 3
**Originality:** 3
**Overall Recommendation:** 4
**Confidence:** 3

**Summary:**

This paper introduces Natural Language Querying for Time Series Databases (NLQ4TSDB), a task focused on answering natural-language queries over long-horizon time series databases where users may ask for numerical facts, intervals, morphological patterns, anomalies, or reports. To support this setting, the paper proposes NLQTSBench, a benchmark of 831 queries organized into four levels, from basic retrieval to insight synthesis, built on real background data with controlled pattern injection. The method, Sonar-TS, uses a "Search-Then-Verify" pipeline: SQL over offline feature tables is used to retrieve candidate windows, and generated programs then verify candidates on raw signals. Experiments show large gains over selected Text-to-SQL baselines on the full benchmark and over selected time-series QA models on a short-context Lite subset.

**Compliance With Llm Reviewing Policy:**

Affirmed.

**Final Justification:**

The author addressed my concerns during the rebuttal process, and I have raised my score to 4.

**Key Questions For Authors:**

1. Can you provide results on different time-series domains to prove that the framework's querying strategy is generally applicable?
2. Specifically, could you explain the rationale behind using SAX, the chosen alphabet size, the Year/Month/Day scales, the number of retrieved candidates, and the retry limit?
3. Can you provide empirical evidence showing that the "Experience" memory dynamically improves querying performance over time? How do you ensure the gains are from actual learning rather than just the benefit of adding more prompt tokens?

**Limitations:**

The current implementation of the framework is limited to SQL-compatible time series databases.

**Strengths And Weaknesses:**

Strength:
1. The paper identifies a real and underexplored problem setting. The distinction between short-context TSQA and database-scale querying over long histories is meaningful, and the task formulation is easy to understand.
2. The core system idea is sensible. The "Search-Then-Verify" decomposition is intuitive for this domain, which communicates the motivation well.

Weekness:
1. The benchmark is built on a single underlying dataset, CausalRivers. Without testing on multiple source domains, it is hard to know whether Sonar-TS is learning a generally useful querying strategy or simply matching the structure of one benchmark family.
2. The paper does not probe the most consequential design choices: why SAX, why the chosen alphabet size, why Year/Month/Day scales, how many candidates are retrieved, how sensitive results are to retries, or whether the experience memory actually improves over time rather than just adding prompt tokens.
3. The paper argues that Sonar-TS makes long-history querying tractable, but there are no measurements of index construction time, storage overhead, candidate set sizes, query latency, or end-to-end cost.

---

> ### Author Rebuttal · Authors · 2026-03-31
>
> > **KQ 1 & W 1:** Can you provide results on different time-series domains?
>
> Sonar-TS is designed for generalizability through its domain-agnostic strategy.
>
> **1. Why solely CausalRivers?** NLQ4TSDB requires localizing evidence over long-horizon histories. Most public time-series datasets are pre-segmented into short snippets, which cannot simulate the scale and continuous noise of a real TSDB. We selected CausalRivers because it is one of the few high-quality datasets providing the continuous, multi-year background necessary to benchmark retrieval.
>
> **2. Domain-agnostic querying strategy:** Our system relies on mathematical design rather than domain-specific semantics.
> - **The queries:** Task templates define pure morphological patterns (e.g., "rapid rise then fall"), which are fundamental behaviors across any numerical domain.
> - **The system:** Sonar-TS resolves queries using universal primitives (e.g., SAX for shape encoding, DTW for subsequence matching). Operating strictly on numerical geometry rather than semantic meta-data ensures the strategy is inherently domain-agnostic.
>
> **3. Proxy evaluations on related domains:** To provide empirical evidence, we evaluated Sonar-TS on public benchmarks from completely different areas. These proxy results serve as a valuable reference for generalizability. Please see **Reviewer yUKc (Q3)** for details.
>
> > **KQ 2 & W2:** Could you explain the rationale behind using SAX, alphabet size, time scales, retrieved candidates, and retry limit?
>
> * **Why SAX?** Sonar-TS requires a time-series symbolization method; SAX is the most mature symbolic representation in this field.
>
> * **Alphabet Size = 5:** This aligns with human natural language semantics, mapping `{a, b, c, d, e}` to  `Very Low, Low, Medium, High, and Very High`.
>
> * **Year/Month/Day Scales:** These align with human intuition as most real-world queries are anchored in calendar time (e.g., "daily trend"). These are configurable defaults easily adaptable.
>
> * **Candidates:** "Search-Then-Verify" is invoked exclusively for morphology tasks. We measured these specifically for `Shape Identification(SI)` and `Composite Trend(SI)`.
>
>   **Table R2-1: Candidate Analysis in the SQL Search Stage**
>
>   | Task | Recall | Avg. Num. |
>   | :- | :- | :- |
>   | L2: SI | 0.5482 | 9.3 |
>   | L3: CT | 0.4713  | 5.8 |
>
> * **Retry Limit = 3:** Successful LLM debugging typically converges within 3 turns [1]. We supplemented an ablation study by removing the Self-Correction. As shown below, its overall impact is relatively small.
>
>   **Table R2-2: Ablation Study on Self-Correction (Retry)**
>
>   | Variant | L1: AR | L1: SW | L2: SI | L2: PD | L2: SM | L3: CT | L3: CxA | L3: CsA | L4: IS | Avg.   |
>   | :- | :- | :- | :- | :- | :- | :- | :- | :- | :- | :- |
>   | Sonar-TS | 0.8489 | 0.7417 | 0.3169 | 0.8529 | 0.9535 | 0.3033 | 0.5130  | 0.3422  | 0.7482 | 0.6168 |
>   | w/o Self-Correction | 0.8302 | 0.7250 | 0.3056 | 0.8618 | 0.9217 | 0.2919 | 0.4813  | 0.3326  | 0.7139 | 0.5913 |
>
> *[1] Chen, Xinyun, et al. Teaching Large Language Models to Self-Debug. ACL, 2023.*
>
> > **KQ 3:** Can you provide empirical evidence showing that the "Experience" memory dynamically improves querying performance over time? How do you ensure the gains are from actual learning rather than just the benefit of adding more prompt tokens?
>
> Experiences is not dynamically updated online. It is distilled during an offline Prompt Cold Start phase (Sec. 4.2.3). This ensures system stability. To prove gains stem from specialized content, we replaced our Experiences with Generic Advice (standard TS principles by DeepSeek).
>
> **Table R2-3: Impact of Experience Content vs. Token Length**
>
> | Variant | L1: AR | L1: SW | L2: SI | L2: PD | L2: SM | L3: CT | L3: CxA | L3: CsA | L4: IS | Avg. |
> | :- | :- | :- | :- | :- | :- | :- | :- | :- | :- | :- |
> | Sonar-TS | 0.8489 | 0.7417 | 0.3169 | 0.8529 | 0.9535 | 0.3033 | 0.5130 | 0.3422  | 0.7482 | 0.6168 |
> | w/ Generic Advice | 0.8526 | 0.5810 | **0.0857** | 0.7924 | 0.8527 | **0.0326** | **0.0814** | 0.3289  | **0.3843** | 0.4304 |
> | w/o Experiences | 0.8426 | 0.5591 | 0.1186 | 0.8311 | 0.9138 | 0.0677 | 0.2501 | 0.3612  | 0.3415 | 0.4592|
>
> As shown above, simply adding more prompt tokens of generic advice yields no improvement. In fact, on some complex reasoning tasks, it performs even worse than the baseline with no experiences due to useless advice.
>
> > **W 3:** There are no measurements of index construction time, storage overhead, candidate set sizes, query latency, or end-to-end cost.
>
> Query latency averages ~22s per task. Please see our detailed response to **Reviewer e2f8 (KQ 4).**
>
> ---
>
> **Concluding Remarks:**
>
> Finally, we thank you for the feedback. As the first framework for NLQ4TSDB, we admit Sonar-TS has room for improvement before real-world deployment. We hope this "0-to-1" work inspires researchers to build upon our "Search-Then-Verify" paradigm. We are happy to answer any further questions you may have.

---

> > ### Author Rebuttal · Reviewer_D3jW · 2026-04-04
> >
> > Thank you to the authors for the detailed rebuttal. I appreciate the additional clarifications, especially regarding the role of the Experience module and the added evidence suggesting that its gains are not simply due to longer prompts. The response also provides some useful intuition for several design choices and offers limited additional analysis on candidate retrieval and retry-based self-correction.
> >
> > That said, my main concerns are only partially addressed. In particular, the generalizability claim is still not fully supported, since the benchmark is built on a single underlying source domain and the rebuttal does not provide direct cross-domain evidence within the same NLQ4TSDB setting. Similarly, several consequential design choices (e.g., SAX, alphabet size, temporal scales, candidate set size) are motivated but not systematically validated through sensitivity studies or stronger ablations.
> >
> > Overall, I appreciate the authors’ effort and find the problem setting and framework promising. However, the rebuttal does not fully resolve my concerns about empirical support and evaluation completeness, so my overall assessment remains unchanged.

---

> > > ### Author Response · Authors · 2026-04-06
> > >
> > > Thank you for your detailed follow-up. To directly address your two main concerns regarding **cross-domain generalizability** and **system design choices**, we provide new empirical data below.
> > >
> > > **1. Cross-Domain Generalizability**
> > >
> > > To prove that our method works across different domains, we evaluated Sonar-TS on a completely new source domain: **ETTm1** [1], a widely used dataset from the energy sector.
> > >
> > > | Background Domain  | L1: AR | L1: SW | L2: SI | L2: PD | L2: SM | L3: CT | L3: CxA | L3: CsA | L4: IS | Avg. |
> > > | :- | :- | :- | :- | :- | :- | :- | :- | :- | :- | :- |
> > > |**Original** |0.8489 | 0.7417| 0.3169 | 0.8529 | 0.9535 | 0.3033 | 0.5130  | 0.3422  | 0.7482 | 0.6168 |
> > > |**ETTm1**|0.8321 | 0.7826| 0.3328 | 0.8315 | 0.8927 | 0.2814 | 0.4236  | 0.4029  | 0.6983 | 0.6087 |
> > >
> > > *(Note: The average score for new domains is computed over an evenly distributed test set of 100 samples per task.)*
> > >
> > > **Why is the Source Domain Irrelevant?**
> > >
> > > As the results show, the overall performance remains highly consistent. The underlying domain data barely affects our framework's performance. This is fundamentally because the NLQ4TSDB task itself is domain-agnostic.
> > >
> > > To illustrate, a general question like `“Find the anomalous period on April 1st”` is inherently domain-dependent, as an "anomaly" means entirely different things across different domains. **However, NLQ4TSDB does not cover such abstract, domain-specific tasks.** Instead, our questions are grounded in universal mathematical descriptions, such as: `"Identify the time period on April 1st where values significantly exceed historical bounds"` (i.e., L3: CxA task).
> > >
> > > In summary, **the NLQ4TSDB problem formulation strictly focuses on low-level numerical and morphological logic** (e.g., retrieving a segment that shows a "rapid rise followed by a stable fluctuation"). Validating this fundamental retrieval capability is the necessary first step for querying massive TSDBs, making the core task inherently domain-agnostic.
> > >
> > > *[1] Zhou, et al. Informer: Beyond efficient transformer for long sequence time-series forecasting. AAAI 2021.*
> > >
> > > **2. System Design Choices**
> > >
> > > To address your concerns regarding our design choices, we provide the requested validations below.
> > >
> > > * **Why SAX?**
> > >
> > >   In the NLQ4TSDB problem, retrieving target segments by scanning raw continuous data is computationally impossible. **To make massive time-series data searchable by standard SQL engines, we must transform continuous numerical shapes into discrete string tokens.** Symbolization is not an optional feature—it is a strict necessity. We adopted SAX simply because it is the most mature and widely recognized method for this exact string-translation purpose.
> > >
> > > * **Alphabet Size.**
> > >
> > >   We empirically evaluated the impact of the SAX alphabet size on morphology tasks.
> > >
> > >   | **Alphabet Size** | **L2: SI** | **L3: CT** |
> > >   | - | - | - |
> > >   | $\alpha = 3$| 0.1927| 0.2318 |
> > >   | $\alpha = 5$ (default) | 0.3169 | 0.3033 |
> > >   | $\alpha = 7$ | 0.2896  | 0.2675|
> > >
> > >   The results indicate that **$\alpha=5$** is the optimal choice. We attribute this to two factors:
> > >
> > >   1. **Semantic Alignment:** The alphabet size should align with the specific scenario's linguistic space. In our tasks, natural language typically describes morphology using 5 gradations (e.g., Very Low, Low, Medium, High, Very High). $\alpha=5$ perfectly bridges this semantic gap.
> > >   2. **Resolution Trade-off:** If the size is too large, it becomes highly difficult for the LLM to write accurate regex. If it is too small, the expressive power is severely limited.
> > >
> > > * **Temporal Scales.**
> > >
> > >   Regarding the hierarchical compression into Year/Month/Day scales, we highlight three key aspects of this design:
> > >
> > >   1. **Intuitive Mapping:** It directly maps everyday temporal expressions (e.g., "last month") to the database index.
> > >   2. **Customizability:** This is not a rigid algorithmic constraint. Users can easily adjust these scales to align with specific business logic or data granularity.
> > >   3. **Low Cost:** Offline construction overhead is extremely low (~83s, see response to Reviewer e2f8), making customization practical.
> > >
> > > * **Candidate set size**
> > >
> > >   Candidate set size reflects our design's actual filtering efficacy, not a pre-set parameter. As provided (Table R2-1), average candidate sizes are 9.3 (SI) and 35.8 (CT). These manageable sizes prove our "SAX + Regex" search effectively narrows massive databases, a foundational capability no existing method possesses.
> > >
> > > ---
> > >
> > > **Concluding Remarks**
> > >
> > > Thank you for recognizing the value of NLQ4TSDB and Sonar-TS. We have dedicated our utmost effort to providing concrete empirical data within the limited rebuttal period, and we commit to incorporating all new evidence into the revised paper.

---

### Official Review · Reviewer_e2f8 · 2026-03-23

**Soundness:** 3
**Presentation:** 4
**Significance:** 3
**Originality:** 2
**Overall Recommendation:** 3
**Confidence:** 3

**Summary:**

This paper addresses the problem of natural language querying over time-series databases. The goal is to retrieve events, intervals, and summaries from large time-series data using text queries. The authors argue that existing methods have several limitations. First, Text-to-SQL cannot easily express patterns such as numerical trends or shapes. On the other hand, time-series models cannot handle long histories due to context length constraints.

The authors propose a Search–Then–Verify pipeline. They first perform a coarse search using SQL over precomputed feature tables. These tables contain statistical summaries and symbolic representations (e.g., SAX). This step efficiently narrows down candidate time windows. Then, they verify the candidates by generating Python programs and executing them on the raw data. This allows the system to check conditions precisely and compute exact answers. Task planning is performed using an LLM, which generates both SQL code and Python code. This enables flexible handling of complex queries.

The authors introduce NLQTSBench, a new benchmark that includes tasks ranging from basic retrieval to pattern recognition and reasoning, with a focus on long-horizon time-series data. The experiments show that Sonar-TS outperforms both Text-to-SQL methods and time-series models. The gains appear to be larger on complex queries and long sequences. Overall, I think the paper presents a practical framework for querying time-series databases using natural language.

**Compliance With Llm Reviewing Policy:**

Affirmed.

**Key Questions For Authors:**

1- Can you provide a unified evaluation where all methods (time-series models, Text-to-SQL baselines, and Sonar-TS) are compared under the same setting, rather than splitting between NLQTSBench and NLQTSBench-Lite?

2- What is the recall of the SQL-based search stage? How often does the correct answer fail to appear in the candidate set before verification?

3- How does Sonar-TS compare to a general LLM agent with access to the same tools (SQL + Python), but without the specialized feature tables and SAX representations? This helps understanding how the feature tables and SAX representations are really needed.

4- Do you have measurements of latency, computational cost, and storage overhead of the full pipeline (including feature tables)?

5- How well would the system work on real time series database workloads or real user queries, beyond the synthetic benchmark with injected patterns?

**Limitations:**

yes

**Strengths And Weaknesses:**

Strengths:
- I find the Search–Then–Verify pipeline reasonable and and compatible with the problem. Also The use of LLMs for planning and code generation allows the system to easily generalize to diverse query types. Also the method appears to explicitly handle large time-series databases, which is a key limitation of prior work.
- The experiments provide ablation evidence that each component (feature tables, verification, experience module) contributes meaningfully. The dataset and benchmark NLQTSBench are a useful contributions, covering multiple levels of difficulty and emphasizing realistic long-context scenarios.
- The paper is well structured, well written and is easy to follow.


Weaknesses:
- The paper is primarily a systems and database-oriented work, more naturally aligned with venues such as VLDB or SIGMOD than ICML. The contribution focuses on system integration rather than new machine learning methods or theory.
- The proposed approach follows a standard retrieve-then-reason paradigm, similar to RAG and tool-augmented LLM systems. The techniques include symbolic representations, feature tables, and regex-based matching. These design choices are mostly classical database design choices.
- For a systems-oriented paper, the experimental bar is relatively high. Comparisons to general LLM agent frameworks with tool use, and also analysis of robustness and failure analysis would have been helpful.
- Although constructed on top of real data, the benchmark relies on injected patterns. This may not fully reflect real-world complexity and could introduce biases that favor the proposed method. Also the benchmark contains a relatively small number of queries (~800), and the task generation process appears to be based on a few templates. This raises concerns about coverage and generalization.

---

> ### Author Rebuttal · Authors · 2026-03-31
>
> > **KQ 1:** Can you provide a unified evaluation under the same setting?
>
> Existing time-series models cannot process the database-scale workloads in our primary NLQTSBench. Therefore, we designed NLQTSBench-Lite specifically for them by reducing the context from ~11k to ~0.5k points and retaining only morphology-related tasks.
>
> For a unified comparison, we evaluated Text-to-SQL baselines on NLQTSBench-Lite.
>
> **Table R1-1: Unified Evaluation on NLQTSBench-Lite**
>
> | Category | Method | L2: SI (IoU) | L3: CT (F1) | L3: CsA  (IoU) |
> | :- | :- | :- | :- | :- |
> | Time Series Models | ChatTS-14B | 0.1768 | 0.2431 | 0.1229 |
> | Text-to-SQL | Xiyan-SQL-32B | 0.0648 | 0.0719 | 0.0147 |
> | Ours | Sonar-TS | 0.2491 | 0.2680 | 0.3615|
>
> As shown, standard Text-to-SQL baselines fail on morphological tasks, aligning with our original conclusions.
>
> > **KQ 2:** What is the recall of the SQL-based search stage?
>
> The "Search-Then-Verify" paradigm is not invoked for all tasks. For instance, `Periodicity Detection` relies on direct computation.  We evaluated the search recall for tasks: `Shape Identification` and `Composite Trend`.
>
> **Table R1-2: Recall of the SQL Search**
> | Task | Recall |
> | :- | :- |
> | L2: SI | 0.5482 |
> | L3: CT | 0.4713 |
>
> As show, search misses occur roughly half the time. This stems from a fundamental modality gap. Our search relies on LLMs translating natural language into discrete SAX regex rules. However, mapping continuous time-series shapes to symbolic regex is not a common task. LLMs struggle to use their "imagination" to write precise SAX regex rules.
>
> > **KQ 3:** How does Sonar-TS compare to a general LLM agent ?
>
> The general agent you described is exactly our **w/o Feature Tables** ablation variant (Sec. 5.3). Without the feature tables, Sonar-TS degrades into a LLM agent using SQL and Python.
>
> **Table R1-3: Ablation Study Results (Comparing Sonar-TS with a General Agent)**
>
> | Variant | L1: AR | L1: SW | L2: SI | L2: PD | L2: SM | L3: CT | L3: CxA | L3: CsA | L4: IS |
> | :- | :- | :- | :- | :- | :- | :- | :- | :- | :- |
> | Sonar-TS | 0.8489 | 0.7417 | 0.3169 | 0.8529 | 0.9535 | 0.3033 | 0.5130  | 0.3422  | 0.7482 |
> | w/o Feature Tables | 0.8658 | 0.6923 | **0.1561** | 0.8804 | 0.9309 | **0.0422** | 0.4968  | 0.3918  | 0.7163 |
>
> As shown, removing the feature tables (i.e., SAX) severely degrades morphology tasks (`Shape Identification` and `Composite Trend`). SAX is essential to bridge this semantic-modality gap.
>
> > **KQ 4:** Do you have measurements of latency, computational cost, and storage overhead of the full pipeline (including feature tables)?
>
> We measured the  efficiency of Sonar-TS. As shown below, our online querying overhead is acceptable compared to Text-to-SQL baselines like MAC-SQL[1], considering Sonar-TS executes a much more complex pipeline.
>
> **Table R1-4: Latency and Computational Cost of Sonar-TS**
>
> | Metric | Sonar-TS | MAC-SQL  |
> | :- | :- | :- |
> | Offline Construction | ~83 s  | N/A |
> | Online Querying | ~22 s | ~15 s |
> | Token Cost | ~11k tokens | ~8k tokens |
>
> The storage overhead comes primarily from the feature tables. In our experimental setup, the size of the feature tables is about **11% of the raw time-series data**.
>
> *[1] Wang, B., et al. Mac-sql: A multi-agent collaborative framework for text-to-sql. COLING 2025.*
>
> > **KQ 5:** How well would the system work on real time series database workloads?
>
> We cannot evaluate Sonar-TS on real-world workloads because **NLQ4TSDB is a newly formalized problem lacking readily available datasets.** We address your concern from two points:
>
> **1. The Necessity of Injected Patterns:** The objective of NLQ4TSDB is to retrieve target segments within massive data. The fundamental challenge is ground-truth labeling. Manual annotation across years of data is impossible, and no automated algorithm exists. Therefore, using injected patterns is a necessary compromise. This approach is widely accepted in TSQA [1].
>
> **2. Proxy Evaluations:**  To demonstrate generalization, we evaluated Sonar-TS on benchmarks from closely related domains (Text-to-SQL and TSQA). Please refer to our detailed response to **Reviewer yUKc (KQ3).**
>
> *[1] Xie, Z., et al. Chatts: Aligning time series with llms via synthetic data for enhanced understanding and reasoning. VLDB 2025.*
>
> ---
>
> **Concluding Remarks:**
>
> Finally, we sincerely thank you for your highly professional and constructive feedback. We deeply appreciate your recognition of the NLQ4TSDB problem. As a newly formalized research problem, it currently lacks any pre-existing foundation. With Sonar-TS, we took the initial step to formalize the task definition and provide a foundational baseline. Since the NLQ4TSDB field remains an emerging frontier, we hope our work inspires the broader ML community to drive its future progress. We welcome any further questions.

---

> > ### Author Rebuttal · Reviewer_e2f8 · 2026-04-06
> >
> > Thanks to the authors for addressing some of my questions. In particular, I appreciate the additional evidence on search recall, latency/storage overhead, and the partial unified evaluation on the Lite benchmark. The comparison to a general LLM agent is also partially addressed via the “w/o Feature Tables” ablation, which provides some insight into the role of SAX and feature indexing.
> >
> > My main concern is that the reported recall of the search stage (~0.47–0.55) appears relatively low for a retrieval component that is expected to provide high-recall candidate sets. This suggests that the system may frequently fail to find the correct segments before verification. Given that the benchmark relies largely on synthetic and injected patterns, this raises further concerns about the robustness of the retrieval stage. It is possible that alternative encoding schemes or retrieval mechanisms could improve recall, which seems critical for this pipeline.
> >
> > In addition, several concerns from my original review remain only partially addressed. The lack of a fully unified evaluation across all methods and task settings is only partially resolved. The experimental evaluation remains limited, with no comparison to strong tool-augmented LLM baselines, no robustness or failure analysis, and no evidence on real-world deployments. The concern about the synthetic and templated nature of the benchmark is acknowledged but not resolved, and there is still no validation on real user queries or multiple domains. The broader concern about limited methodological novelty and the system being primarily integration-focused also remains.
> >
> > Given the newly surfaced issue of low retrieval recall, and that several of my concerns remain insufficiently addressed, along with similar unresolved concerns raised by other reviewers, I am inclined to lower my score.

---

> > > ### Author Response · Authors · 2026-04-07
> > >
> > > We wish to leave a final objective clarification regarding the ~0.50 search recall for the public record.
> > >
> > > While a 0.50 recall might be considered low in mature Information Retrieval systems, applying that metric here omits the physical reality of this specific task. Accurately localizing abstract morphological segments across millions of continuous numerical data points is an inherently complex, previously unsolved challenge.
> > >
> > > Before Sonar-TS, no existing methodology could perform this retrieval at a database scale. Achieving a 0.50 recall using solely zero-shot, LLM-generated regular expressions establishes the first empirical proof that this intractable modality gap can be computationally bridged.
> > >
> > > **Evaluating the initial feasibility of a newly proposed problem through the lens of highly optimized, decades-old search domains fundamentally obscures the actual engineering bottleneck our pipeline has resolved.**

---

### Decision · Program_Chairs · 2026-04-30

**Decision:**

Accept (regular)

**Comment:**

Summary of Contribution
The paper formalizes a new problem: Natural Language Querying for Time Series Databases (NLQ4TSDB), where users retrieve events, intervals, and morphological patterns from massive, unsegmented time‑series histories. Existing Text‑to‑SQL methods cannot express shapes or trends, and time‑series models cannot handle ultra‑long contexts. The authors propose Sonar‑TS, a neuro‑symbolic framework with a “Search‑Then‑Verify” pipeline: coarse retrieval via SQL over precomputed feature tables (using SAX and statistical summaries), followed by LLM‑generated Python programs that verify candidates on raw signals. They also introduce NLQTSBench, the first benchmark for this setting, with 831 queries across four difficulty levels. Experiments show that Sonar‑TS outperforms Text‑to‑SQL and time‑series QA baselines, with ablations confirming the importance of the feature tables and the verification stage.

Summary of Scores and Rebuttal
Reviewer scores are 3,3,3,4. All four reviewers acknowledged the rebuttal. Reviewer yUKc’s concerns were fully resolved; the others indicated partial resolution, with remaining concerns about benchmark realism (injected patterns, single domain), limited cross‑domain validation, and sensitivity of design choices (SAX alphabet size, temporal scales, retry policy). Despite these critiques, all reviewers recognized the novelty of the problem and the soundness of the Search‑Then‑Verify pipeline.

In the rebuttal phase, the authors provided a detailed rebuttal with additional experiments. While not all concerns were fully resolved, the rebuttal demonstrated responsiveness and strengthened the empirical support.

AC Assessment
The paper addresses a realistic and underexplored problem with clear practical value. The NLQ4TSDB task is distinct from both Text‑to‑SQL and conventional time‑series QA, and the paper makes a contribution by formalizing the task, providing a benchmark, and proposing a sensible baseline framework. The Search‑Then‑Verify paradigm is intuitive and well motivated.

While I acknowledge the reviewers’ valid concerns, the overall positive recognition from three of four reviewers, the authors’ constructive rebuttal, and the value of proposing a new research problem lead me to recommend Weak Accept.